# The Methodology of Virtualizing Sculptures and Drawings: A Case Study of the Virtual Depot of the Gallery of Matica Srpska

Miloš Obradović [1], Snežana Mišić [2], Ivana Vasiljević [3],*, Dragan Ivetić [4] and Ratko Obradović [3],*

1   Department of Architecture and Urban Planning, Faculty of Technical Sciences, University of Novi Sad, 21102 Novi Sad, Serbia; milos_obradovic@uns.ac.rs
2   The Gallery of Matica Srpska, 21101 Novi Sad, Serbia; s.misic@galerijamaticesrpske.rs
3   Computer Graphics Chair, Faculty of Technical Sciences, University of Novi Sad, 21102 Novi Sad, Serbia
4   Chair of Applied Computer Science, Faculty of Technical Sciences, University of Novi Sad, 21102 Novi Sad, Serbia; ivetic@uns.ac.rs
*   Correspondence: ivanav@uns.ac.rs (I.V.); obrad_r@uns.ac.rs (R.O.); Tel.: +381-21485-2297 (R.O.)

**Abstract:** The aim of this paper is to introduce the public to the virtual depot of the Gallery of Matica Srpska (GMS), which showcases a collection of sculptures that have been digitized using a Structure-from-Motion photogrammetry and presented by the use of virtual reality and paintings, and drawings were presented through a digital flipbook application. Through the application of cutting-edge methods, highly precise digital replicas of these significant cultural artifacts have been created with details that are difficult to see without a magnifying glass when observing the physical artifact. Additionally, it explores the profound significance and advantages of the virtual depot, such as facilitating remote access, augmenting preservation endeavors, and fostering interdisciplinary collaborations, academic research, educational purposes, and public engagement. The implementation of the virtual depot offers a novel approach to showcasing and studying cultural heritage, opening up new possibilities for the exploration and appreciation of these artifacts in a digital environment. Today, the collection of the GMS encompasses more than 10,000 art objects, which means that one visitor would need about 53 years to access each individual artifact. Virtual depot enables each visitor to do this much faster, but more importantly, in any occasion and setting that they find suitable.

**Keywords:** virtual depot; photogrammetry; cultural heritage; preservation; public engagement; virtual reality; novel approach; sculptures; drawings

## 1. Introduction

The Gallery of Matica Srpska (GMS) [1], which boasts a unique collection of Serbian art from the end of the 16th century to the present day, is the oldest and richest national art museum in Serbia. It was founded in 1847 in Pest (Hungary) under the auspices of Matica Srpska, the oldest literary, cultural, and scientific society among the Serbs. After Matica Srpska was moved to Novi Sad in 1864, the art collection assembled by that time was transferred too and was shown to the public for the first time in 1933. In the following decades, the collection was significantly increased by gifts, bequests, and purchases, which was particularly intensified at the beginning of the 21st century.

The diverse chronological, thematic, and technical collection of the Gallery of Matica Srpska, which today numbers more than 10,000 art objects, consists of collections of paintings, drawings, graphics, sculptures, combined techniques, applied art, decoration, numismatics, seals, copies of stampers, clichés, and new media. It features masterpieces by prominent artists such as Hristofor Žefarović, Teodor Kračun, Katarina Ivanović, Pavle Simić, Đura Jakšić, Uroš Predić, Paja Jovanović, Đorđe Jovanović, Sava Šumanović, Milan Konjović, Petar Lubarda and many others.

The 175-year history of the Gallery of Matica Srpska's collection is also a partial history of the Serbian people, their art, and their culture. It reflects all the socio-historical, cultural,

and artistic stages of the Serbian society, which inhabited a wider geographical area—from the former Austrian and Ottoman empires through the Principality, that is, the Kingdom of Serbia, various forms of the state organization of the former Yugoslavia, to the present-day Republic of Serbia. The collection of the Gallery of Matica Srpska best illustrates the origin, development, and achievements of Serbian art and points to the origins, position, and place of Serbian culture and art in the European context.

Today, the Gallery of Matica Srpska is recognized as a place of intertwining European and national art, innovative approaches and attractive themes, top production, a place of aesthetic experience, dialogue, learning, creative thinking, and unique experience.

Aware of the fact that a modern approach to the development of museum practice is unthinkable without the application and use of modern information technologies in informing about cultural heritage, its availability, presentation, and promotion, the Gallery continuously works on introducing new, modern procedures and techniques for documentation, processing, publishing, as well as for the presentation of works of art [2].

GMS continued to develop and implement activities in the field of digitization through the project "Digitization and virtual presentation of the art collection of the Gallery of Matica Srpska", which is realized in cooperation with the Faculty of Technical Sciences of the University of Novi Sad—Computer Graphics Chair (The cooperation between the Gallery of Matica Srpska and the Faculty of Technical Sciences of the University of Novi Sad was formalized by the signing of the Agreement on Cooperation (2019–2023), which includes the joint implementation of projects in the field of digitization of cultural heritage, work on the introduction of new technologies into the operations of the Gallery, etc.). This project is in accordance with the 2017–2027 Strategy of the Ministry of Culture of the Republic of Serbia, which, among other things, recognizes the digitization of cultural heritage as an important factor in the preventive protection of Serbian cultural heritage. This ensures greater visibility, better presentation, and promotion of the national cultural heritage [3].

With the development of technologies and digital tools for the presentation of architectural objects in recent years, a synergy of virtual reality and photogrammetry has been developed as a tool for an increasingly widespread way of visualizing, presenting, preserving, and spreading cultural heritage. Photogrammetry offers a reliable method for realistic and accurate modeling of objects based on photographs of objects from the real world, while virtual reality provides not only visualization but also an immersive and interactive experience of a photogrammetrically reconstructed object of cultural heritage. The process of preserving cultural heritage is reflected in the digital formatting of certain objects, which users can visualize using a computer and adequate equipment. In this way, they can observe the object from different angles, which may not be possible during live observation as cultural monuments and heritage can be destroyed or inaccessible to the public. In addition to the given possibilities, by using the game engine for interactive visualization, it is possible to present the architectural cultural heritage in a way that is difficult, or impossible, to present in 3D space. This may spur further interest in learning about cultural heritage.

United Nations Educational, Scientific and Cultural Organization (UNESCO) (UNESCO. https://uis.unesco.org/en/glossary-term/cultural-heritage, accessed on 28 August 2023) defines "cultural heritage" as the legacy of artifacts, monuments, a group of buildings and sites, and museums with different values including symbolic, historical, artistic, aesthetic, ethnological, scientific, and social significance. Cultural heritage includes tangible heritage (movable, immobile, and underwater) and intangible cultural heritage (ICH) built into cultural and natural heritage artifacts, sites, or monuments. Intangible cultural heritage does not include other cultural domains such as festivals, celebrations, etc., but covers industrial heritage and cave paintings.

The example of the reconstruction of a square in Rome shows the application of interactive technologies in the field of cultural heritage. This represents a didactic tool because it improves cognitive processes by making historical and archaeological data easily understandable to everyone, and within video games, this potential is strengthened by the

dynamics of learning by doing. However, there are limitations, including a great effort required to ensure the consistency and reliability of the reconstruction, as well as the ability of people to deal with both computer graphics and archaeological matters. So, it is the video game that has to match and adapt to the historically validated 3D environment in order to maintain consistency [4,5]. The design of a VR application for cultural heritage requires various professional skills and presents some complexity in coordination and management. This paper presents strategies to overcome these problems, suggesting guidelines for the development of VR systems for cultural heritage. It illustrates a complete methodology for the creation of a virtual exhibition system based on realistic, high-quality 3D models obtained from archaeological excavations (reconstructed using a 3D scanner and a high-definition camera) and a low-cost multimedia stereoscopic system called MNEME, which allows the user to freely and easily interact with the rich collection of archaeological findings. It has been shown that with a relatively small budget, it is possible to recreate and display a rich archeological collection in an adequate way [6].

Mah et al. [7] showed how the Tampines Chinese Temple in Singapore was used as a detailed methodological framework for creating a virtual tour in order to preserve the physical built environment, as well as intangible historical and sociocultural elements in the space of the cultural heritage site. The data to create the virtual tour were collected using a 360° degree camera and focused on facilitating the creation of future applications of a similar type. Sala [8] shows in his research the examples of the development of virtual reality within architectural and engineering education and gives conclusions in the form of more interactive lectures and multimedia as a means of helping the teaching process. Also, the benefits of VR are listed, which are seen in the form of overcoming language barriers, modeling, design, marketing, and the like. The project by Nisiotis et al. [9] seeks to develop new methods that will relate to museum visitors, influenced by modern technologies such as social media, smartphones, the Internet, smart devices, and visual games, providing a unique experience of exploring and interacting with real and virtual worlds simultaneously. The focus of Monaco et al.'s research [10] is to leave the role of curator of the exhibition to the lovers of cultural heritage. When creating virtual exhibits, users must perform a data selection phase that represents several stages, including finding data sources and extracting data of interest. Users can search for geographically distributed artworks thanks to their associative nature, manipulate heterogeneous data, and easily customize their exhibitions using the wide range of cultural heritage graphs available. In research [11], the state-of-the-art systems of augmented, virtual, and mixed reality are investigated by Bekele et al. as a whole and from the perspective of cultural heritage. Specific areas of application in digital cultural heritage are identified, and suggestions are made as to which technology is most appropriate in which case. In [12], Maiwald et al. analyzed the idea of the existence of a virtual time machine in a digital environment such as virtual reality or four-dimensional (4D) geographic information systems, which require accurate positions and orientation of objects from historical images. Texturing of 3D models is enabled, and new perspectives of spatial distributions of historical data are offered, as these photographs are sometimes the only visual remains of objects due to the reconstruction or destruction of the object in question. The challenge and subject of research in this paper are reflected in the assessment of the pose of objects, as well as in the retrieval of photographs.

The paper by Rahaman et al. [13] aims to be helpful for less expert but enthusiastic users by providing aid to create image-based 3D models and share them online, as well as to enable the audience to use 3D models in the real world, using mixed reality. Arrighi et al. [14] emphasized visualization of the Victoria Theater in Newcastle, Australia, using virtual reality to preserve the site's heritage. The traditional way of moving in 6 degrees of freedom VR application relies heavily on the teleportation of users moving around the virtual space. However, this can be physically and mentally challenging for some new VR users and users with limited mobility, and emphasis is placed on other modes of movement, which could improve the perception of VR and simplify its use. In the field of architecture, the use of virtual reconstructions of historical sites has played a

significant role in the advancement of digital technologies, including digital surveying, virtual reality, and augmented reality [15–17]. Integrating serious game technology within museum environments has opened up new possibilities for educational and entertaining museum experiences. This contribution outlines the principles and guidelines that shaped the digital museological design process of the Virtual Museum EPANASTASIS-1821 [18].

Virtual reality (VR) and information and communication technology (ICT) have opened up new possibilities for understanding and promoting cultural heritage within the museum context. These technologies allow users to experience cultural heritage without physically interacting with real objects. For museum institutions, VR is a valuable tool for performing various cultural tasks and effectively engaging with the public [19]. The cultural heritage sector increasingly integrates augmented and virtual reality (VR) solutions to meet dissemination and interpretation needs for its collections. As research in the field grows, the required entertainment and learning impacts of such applications are rising. Tsita et al. [20] present a VR museum that aims to facilitate an understanding of cultural heritage. The utilization of augmented reality (AR) and virtual reality (VR) in cultural heritage promotion applications has proven to be effective in enhancing visitors' experiences, engaging existing audiences, and attracting new ones [21,22]. Furthermore, Lee et al. [23] demonstrated that an immersive VR environment can elevate the tour experience and motivate users to explore the physical space. In addition, when evaluating VR applications, the focus is typically on visitors' experiences.

However, as noted by Shehade and Stylianou-Lambert [24], limited consideration is often given to the perspectives of museum professionals regarding the optimization of design and development for such applications. The collaboration between museums and public bodies has facilitated increased visibility and attracted attention, leading to a significant expansion of museums' online presence. This has been achieved through the creation of new websites and the establishment of official profiles on major social networks [25–27]. In recent times, virtually all museums have recognized the importance of digital communication in promoting their institutions and engaging with an increasingly online audience, blurring the boundaries between the physical and digital realms [28–30]. Simultaneously, digital technologies employed in virtual representations of built heritage offer researchers and scholars a wide array of possibilities [31].

Over time, methodologies and operational practices have been developed in architecture, restoration, and archaeology, drawing upon the theoretical and methodological foundations of these disciplines. Internationally, numerous research groups have adopted diverse and intriguing approaches, reflecting various scientific and cultural backgrounds [32–34]. With the advancement of acquisition, analysis, and interactive digital representation methodologies and tools, it has become feasible to explain and convey the contents of museum collections through virtual and augmented reality (VR-AR) environments [35,36].

The advent of technology has ushered in transformative changes in the cultural heritage sector, with a growing focus on virtual museums and the digitization of cultural artifacts. These types of projects are aimed at preserving and presenting cultural heritage in innovative ways, but they can vary significantly in terms of equipment, budget, and the number of individuals involved [37]. Countries such as the Czech Republic, Finland, Greece, Latvia, Lithuania, Poland, Slovakia, and Sweden are examples that have used the European Structural and Investment Fund to co-finance cultural heritage digitization. The Lithuanian e-Paveldas project was financed in such a way. The budget of the project was 3.6 million EUR and lasted two and half years. Created a database containing three million pages of books, newspapers, artworks, manuscripts, and church registers. During the project, it was created 32 jobs [38]. In Italia 2008, the Cultura Italia portal was launched. The main aim of this portal is to promote Italian cultural heritage and to provide online access to cultural content. It included 2.5 million items coming from 32 different partners, public and private. Also, a National Archivist System was created [39]. The EU's South East Europe Transnational Cooperation Programme 2011–2013 funded the project entitled "Achieving

SUSTainability through an Integrated Approach to the Management of CULTural Heritage", coordinated by the City of Venice involved 12 institutions from seven different countries, including Romania [40]. There are more examples of EU-funded projects on cultural content digitization: RICHES (Renewal, Innovation and Change: Heritage and European Society), PREFORMA (PREservation FORMAts for culture information/e-archives), Civic EPISTEMOLOGIES [41]. In the Republic of Serbia, there are guidelines from the Ministry of Culture and Information for implementing the digitization of cultural heritage, but a common information system in the field of culture in Serbia, which is managed by the Government through state administration bodies and in cooperation with cultural institutions, has not yet been fully established [42].

The budget of the project, which is the subject of this paper, is about 500 times smaller than the Lithuanian e-Paveldas project. Twelve people were hired, eight of whom worked on the digitization and virtual presentation of the depot of drawings and graphics, and it lasted one year. This fact is an explanation of why there digitized seven sculptures and 33 drawings, but also the basic difference of the implemented project in relation to the previously mentioned and similar ones, which were not mentioned.

### 1.1. Photogrammetry

Photogrammetry is a prominent technique for reconstructing objects of cultural importance [42–44]. It involves creating the geometry of a 3D model based on images. This means that software specializing in photogrammetric reconstruction automatically creates a 3D model based on the data it receives from a set of images. This way of obtaining a model is faster than traditional manual 3D modeling procedures based on modeling by using reference images showing orthogonal projections of the object. In addition, the textures obtained by photogrammetric modeling are realistic (the information is obtained from the photographs of the objects being reconstructed) and do not represent the creation of an "imitation" texture based on a reference image [45,46].

Structure-from-Motion (SfM) type of photogrammetry surveying implies a close-range photogrammetry approach, which is based on triangulation, i.e., determining the coordinates of the points in space in relation to the reference coordinate system is performed based on the position of the points recorded on two adjacent photographs [47,48]. SfM is suitable for producing precise and detailed 3D models that are useful for different types of analysis, visualization, and documentation [49–51].

To ensure the quality of the 3D model obtained by photogrammetric reconstruction, it is crucial to create a set of photographs based on the detailed geodetic survey plan [48,52]. A well-planned surveying guarantees a set of photographs that is usable in the sense that the photos are of high quality in terms of color, texture, and dimensions of the object, that they are not blurred, and that the recording covers all parts of the object. Today, there is a variety of image-based modeling software available, both open-source and commercial [46,51,53,54].

### 1.2. Virtual Reality for Creating Virtual Museums

Virtual reality (VR) applications provide users with a completely novel experience and absolute immersion into a virtual space [55]. Although the first experiments with VR head-mounted displays were carried out in 1965, their use in education was not tested until 1990 [56]. The rise in popularity of video games and the creation of the Oculus Rift VR headset, as well as the creation of Google Cardboard, influenced VR applications to become the main type of presentation today [57].

Nowadays, museums are increasingly using VR applications as an innovative way to entertain audiences and inform them about their activities [58,59]. The advances in technology and the development of modern techniques for creating virtual museums, such as VR, enable the revival of history and the presentation of cultural heritage in a way that was unimaginable until recently. This type of presentation is interesting and enlightening for all generations, even for the youngest [60]. The presentation in virtual reality transcends

space and time, i.e., it allows users to visit distant places, which in real life they might never visit, but they can also visit them in another period and see how they once looked [33]. Taking into account all these facts, VR is an excellent choice for the presentation of the depot of the Gallery of Matica Srpska in Novi Sad in order to make it accessible to the public, at least in the virtual world. That is the reason why this technique was used in this paper.

The coronavirus pandemic and the global shutdown contributed to the development and creation of an increasing number of virtual galleries and museums, and VR applications for promoting tourism [61–64].

Lutreh et al. [65] created an overview of various types of virtual museums as native artifacts or as digital twins of physical museums. The authors conclude that it is necessary to digitize and present cultural heritage with the cooperation of all interested groups, according to uniform norms and standards, and respecting digital property rights. Pisoni et al. [66] have also made a review of the literature on how artificial intelligence (AI) is used for designing accessible cultural heritage. The authors have created a conceptual framework in which they show what are the key elements that make up the online experiences of museums and cultural heritage and how these elements are interconnected. They presented a Conceptual framework for AI-enabled accessibility of museums and cultural heritage sites. This framework includes more elements such as the concepts of learning progress in curiosity-driven exploration [67], pedagogical agents [68], creative AI, natural language explanation generation, accessible Cultural Heritage through explainable AI [69], and automatic curriculum learning. Lucchi et al. [70] presented a systems approach to creating positive interactions between built, human, and natural systems, whose purpose is to enhance the heritage and the natural environment, human life, social equity, and economic sustainability. This paper presents the workflow of multi-criteria decision analysis for the regenerative design of archaeological sites, and the method is applied in the Italian site of Casterseprio to verify its feasibility for mapping the interaction between heritage, environmental, social, and economic dynamics in a real case study.

The rest of the paper is organized in the following way. In the second section, a case study of the Gallery of the Matica Srpska is presented. In Section 3, there is presented materials and methods used for project implementation. In Section 4, the obtained results are discussed. Then, Section 5 concludes with suggestions for future research. In Section 6, the authors suggest how to protect their results.

## 2. A Case Study: The Depot of the Gallery of Matica Srpska

The project "Digitization and virtual presentation of the art collection of the Gallery of Matica Srpska", which started in 2021, is aimed at the digitization and virtual presentation of museum objects that are often physically inaccessible to visitors to the Gallery and the general public because it is impossible to exhibit them all within the permanent exhibition and/or on occasional exhibitions. Unlike the traditional way of exhibiting works of art within the physical space of the exhibition, digital and virtual presentations provide numerous opportunities for detailed and comprehensive visualization of objects that cannot be fully exhibited or are in storage and, therefore, rarely available to the public. In this way, the existing limitations and obstacles to the audience's encounter with works of art are overcome, and the method of preservation and protection of cultural heritage objects is improved. In this paper, the aim is not placed on the virtual presentation because the depot itself is not used for exhibitions. A virtual depot differs from a virtual collection because it is a virtual representation of a real depot with works of art that are located there, stored in specific conditions, and not accessible to the public.

The first phase of the project involved the digitization of two works of art painted on paper that had not previously been exhibited and were not available to the public. The first work is an album with graphics and drawings by 18th- and 19th-century artists (24 graphics–copper engravings and 5 drawings), and the second is a 1921 sketchbook by our renowned artist Sava Šumanović containing 14 of his drawings. In addition, a

virtual depot of the Gallery was created. This idea reflects the desire to demystify this "hidden" part of the Gallery and "reveal" it to the public, given that access to the depot and handling of museum objects is restricted to professionals—curators handling collections and conservators–restorers (Professional instruction on conditions and methods of preservation and use of artistic-historical works, National Museum in Belgrade, number: 198/4 dated 11 June 2001), (Rules on conditions, storage, and use of museum objects in the Gallery of Matica Srpska, arch. no. 1103-90/1 of 12 August 2021). The depot is a specially designated and arranged space with an alarm system and access control in accordance with the norms established by the museological profession, which has the exclusive purpose of housing and storing museum objects [71]. It is arranged in such a way as to provide a safe and inspected accommodation of museum objects depending on the type of material (panels, racks, shelves, cupboards, tools…) and the conditions for their safe storage by regular control of microclimatic conditions: prescribed range of air temperature (15–24 °C), the percentage of relative humidity (45–65%) and light depending on the type of museum objects stored in it, as well as preventing other hazards such as fire, unfavorable temperatures, water spills or other harmful effects that could destroy the museum objects housed in it [72]. Through the 3D digitization of the representative objects from the depot, which is achieved using the digital photogrammetry method, and the creation of a virtual space of the depot with a display of the digitized objects (paintings, graphics, drawings, sculptures, works of applied art), insight was provided into the contents of the original gallery depot. This provides the users with the opportunity to familiarize themselves with the Gallery's virtual depot and to see the ways in which its works of art are stored and preserved. Striving for innovative approaches to exhibiting and presenting national fine art, as well as a professional attitude toward the care and protection of its artwork, the GMS works intensively to improve the conditions for the preservation and protection of national heritage and modernize approaches of study and presentation of works of artistic-historical and cultural importance. This is enhanced by introducing digitization and attractive ways of virtual presentation in its permanent and occasional exhibitions.

## 3. Materials and Methods

### 3.1. Photogrammetric Surveying of Sculptures from the Depot

The sculptures from the depot were digitized by the use of photogrammetry. Before the start of the photogrammetric surveying, the on-site surveying plan was created in accordance with the real situation on the ground. This meant taking into account the dimensions of all the sculptures being photographed, their texture and material properties, the lighting in the room where the sculptures were photographed, and the level of detail that should be acquired on the 3D virtual model. For that reason, before creating the on-site surveying plan, the team members went to the GMS to see the real situation on the ground and measure every single sculpture. Creating the on-site surveying plan also involves determining the Ground Sample Distance (GSD) value. This value represents the pixel dimension expressed by the detail dimension of the object, which means that if the object is smaller, the GSD value will be smaller, too. The dimension of the smallest detail on the object being surveyed must be taken into account when determining the GSD value, i.e., the GSD value must be less than the dimension of the smallest detail. As for the settings related to the camera, the values for the focal length, distance from the camera to the object, and distance between two camera locations must be precisely defined in the surveying plan.

The surveying plan for this project was created taking into account the specific sculptures' shape and reflective material from which they were made, as well as the interior light conditions. Figure 1 shows some of the sculptures from the GMS depot that have been digitized, with a white canvas in the background that serves to separate the sculpture from its surroundings.

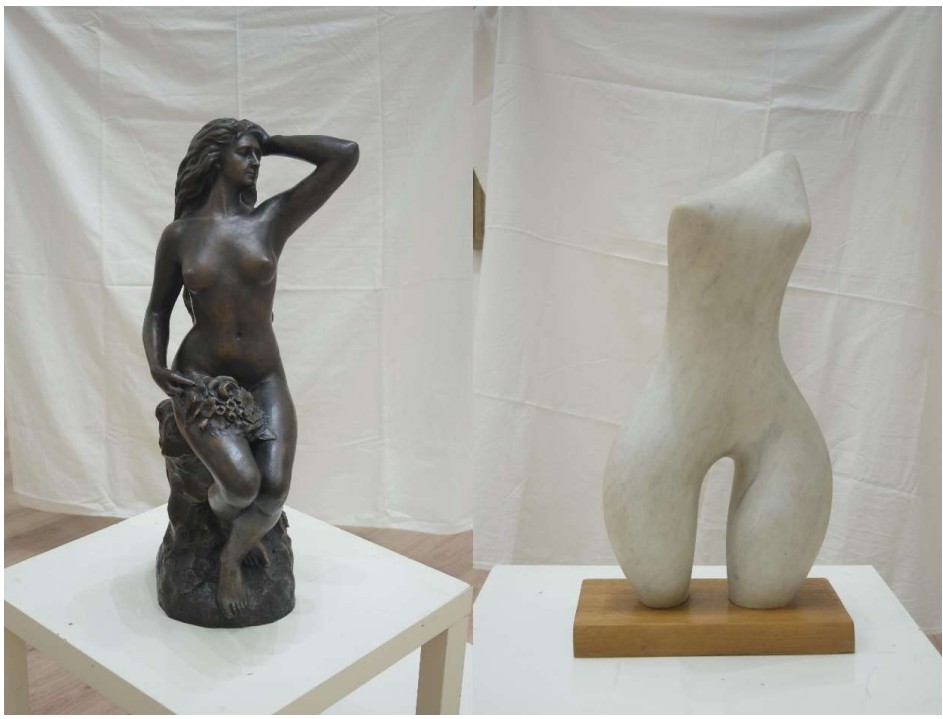

**Figure 1.** Example of two digitized sculptures from the GMS depot.

The equipment used for surveying is a tripod and NIKON D7000 camera with the following characteristics: pixel size—4.78 μm, sensor size—23.6 × 15.6 mm, focal length range—18–109 mm, crop factor—1.53, with manual settings of the following parameters: f-stop, ISO speed, Exposure time.

As the goal was to obtain high-precision 3D models of the sculptures, a GSD value of 0.15 mm was selected. By considering the number of pixels of the camera, it is possible to calculate the covered space (D) in the photos, and it was performed in the following way:

$$D = \text{pixel number} \times \text{GSD} \tag{1}$$

The covered space for portrait mode is 0.1 mm·4928 = 492.8 mm, and for landscape mode is 0.1 mm·3275 = 327.5 mm. In this survey, portrait mode was used. The focal length (c) used is 35 mm. The distance from the camera (whose trajectory is a circle in the center of which the sculpture is located) to the sculpture should be the same on all sides. The scale (m) was calculated as follows:

$$m = \text{GSD}/(\text{pixel size}) = 0.1/(4.78 \times 10^3) = 0.0209 \times 10^{-3} \tag{2}$$

and the distance of the camera from the object (h) is determined by the equation:

$$h = m \times c = 0.0209 \times 10^{-3} \times 35 = 0.735 \text{ m} \tag{3}$$

In order for the sculptures to be photographed from all angles and for the photographs to gather information about the shape of the geometry of the object, it was necessary to photograph the sculptures in 3 stripes. The sculptures were photographed in the hall of the Gallery of the Matica Srpska, and other camera parameters, such as ISO speed, Exposure time, and f-stop, are measured and manually set on-site because their value depends on light conditions on the field. The mentioned parameters must be well adjusted for the reason that the information about the color of the texture must not be changed in relation to the real state. As the sculptures are of similar dimensions and the surveying was performed in the same room with the same light conditions, the created surveying plan was applied to all of them.

The parameters used for photogrammetry surveying are shown in Table 1.

**Table 1.** Parameters used for photogrammetric surveying.

| Photogrammetric Surveying Parameters | All Seven Sculptures |
|---|---|
| Focal length (c) | 35 mm |
| f-stop | f/5.6 |
| ISO speed | 125 |
| Exposure time | 1/8 s |
| GSD | 0.1 mm |
| D | 0.49 m |
| h | 0.73 m |

The data acquired by the photogrammetric surveying process are the images of the photographed sculptures, with output in JPEG and RAW format. An average of 35 images were used to generate the 3D model of the sculpture. The virtual 3D models of the sculptures were generated based on the obtained images in AgiSoft Metashape Professional 1.5.3 software [73], which is automatic software specialized for the photogrammetric reconstruction of objects. The final results of the photogrammetric reconstruction are high-quality 3D models (meshes) of photographed sculptures exported in OBJ file format with a realistic texture saved in PNG file format. In the mentioned formats, 3D models and textures are imported into the game engine for presentation in virtual reality, but after optimization and retopology, procedures have been performed. Figure 2 shows the texture of the sculpture shown in Figure 1 on the left and its 3D model in software.

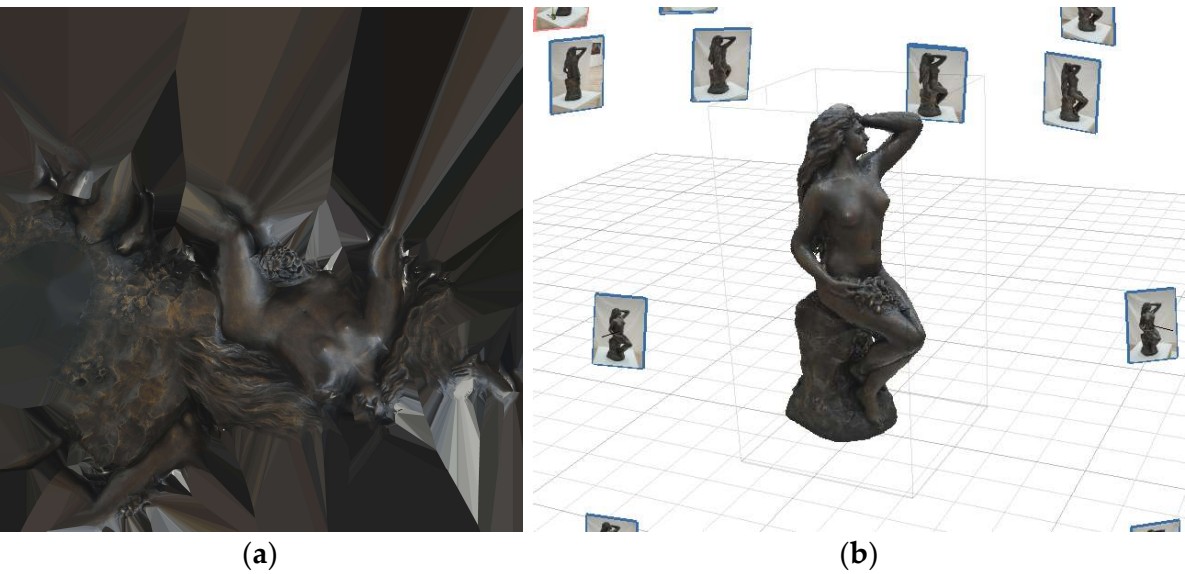

(**a**)                                                                              (**b**)

**Figure 2.** Sculpture shown in Figure 1 on the left: (**a**) an example of the generated texture for digitized sculpture; (**b**) generated 3D model in AgiSoft Metashape with its texture.

As 3D models are created by triangulation, i.e., by finding one point of the model on 2 images by software, it is very important that there is an overlap between 2 adjacent images of about 80%. According to the report that was received after generating one of the sculptures (and the situation is similar with the others), the overlap was rated with the highest rating, which can be seen in Figure 3.

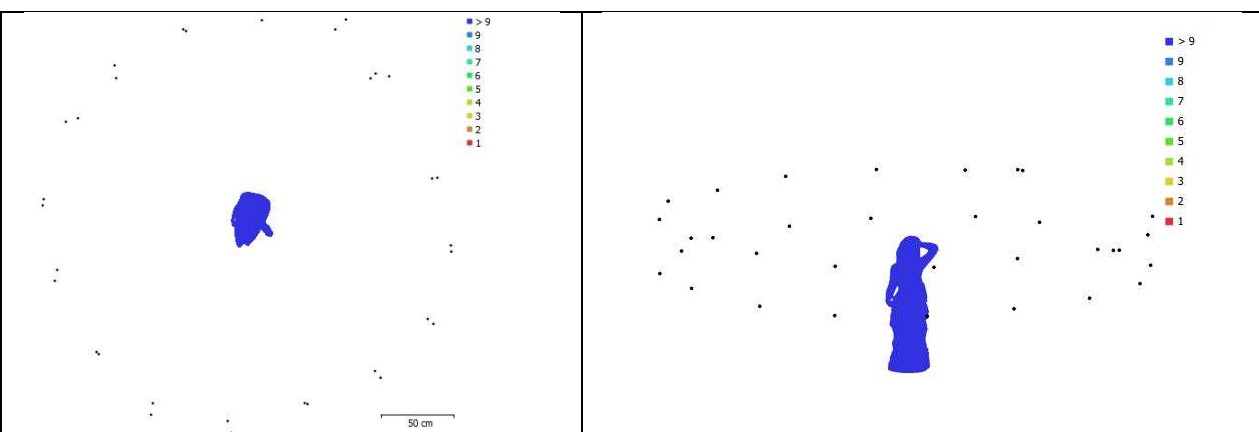

**Figure 3.** Camera overlap for sculpture shown in Figure 1 on the left.

Since the canvas was used as a background during the surveying of the sculptures, and even before the 3D reconstruction, masks were created that separated the object from the background, errors during the reconstruction were reduced to a minimum, i.e., they did not occur. This is proven by the high-quality 3D models with textures that are generated. Also, the precisely created surveying plan contributed to good results.

In the AgiSoft Metashape software, there is an option named Estimate Image Quality; all photos have this parameter higher than the recommended 0.5. and this value ranges from 0.77 to 1.43. This assessment is not perfect, especially where the object has a surface that makes it difficult for the software to identify sharpness. Table 2 shows the RMS and maximal reprojection error of the generated 3D models.

**Table 2.** Generated errors in 3D sculpture models (RMS and max reprojection error).

| Sculpture | RMS Reprojection Error (pix) | Max Reprojection Error (pix) |
|---|---|---|
| Sculpture 1 | 0.135289 (0.42536 pix) | 0.672954 (9.12578 pix) |
| Sculpture 2 | 0.163611 (1.03643 pix) | 0.766344 (23.2326 pix) |
| Sculpture 3 | 0.183698 (0.583207 pix) | 0.563625 (7.98763 pix) |
| Sculpture 4 | 0.157033 (0.419961 pix) | 0.474249 (7.45644 pix) |
| Sculpture 5 | 0.115555 (0.492735 pix) | 0.347808 (10.6593 pix) |
| Sculpture 6 | 0.151696 (0.483613 pix) | 0.455113 (7.29774 pix) |
| Sculpture 7 | 0.114306 (0.520843 pix) | 0.350654 (18.1461 pix) |

The 3D models obtained by photogrammetric reconstruction are of high quality, which means that in addition to the large number of details they contain, they are made up of a large number of polygons. For the needs of virtual presentation in virtual reality, 3D models were optimized [74]. This means that the number of polygons has been reduced while maintaining the level of detail. Also, the polygons that were triangles were turned into quadrilaterals. Zbrush [75] software was used for this purpose.

Figure 4 shows high-quality 3D models of the two sculptures shown in Figure 1.

The 3D models of the digitized sculptures were scaled within the Agisoft Metashape software by introducing control points into the generated 3D reconstruction. Control points are dimensions on a ruler scale that are photographed together with the sculptures in several photos, which allows the introduction of a realistic measurement into the software (see Figure 5).

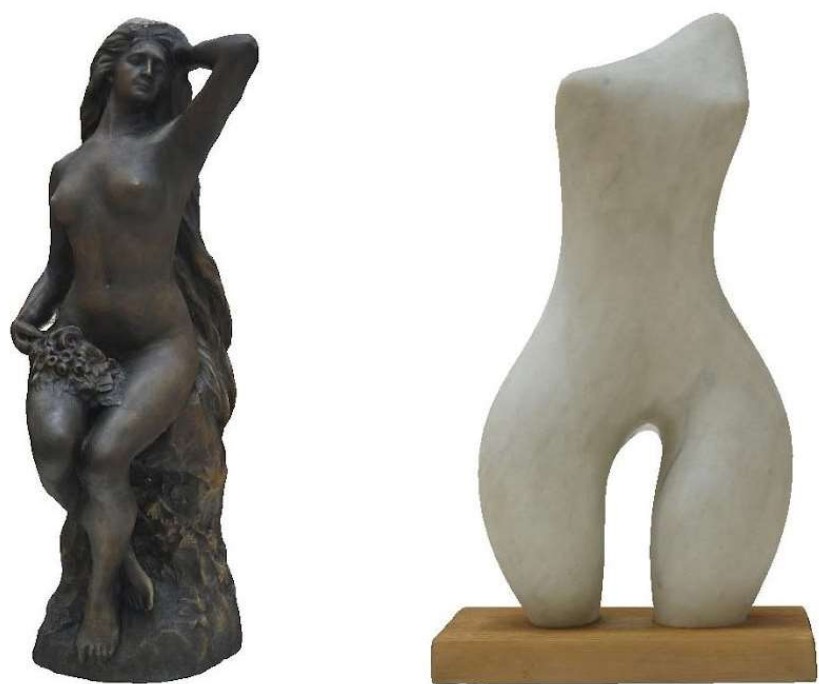

**Figure 4.** High-quality 3D models of the two sculptures shown in Figure 1.

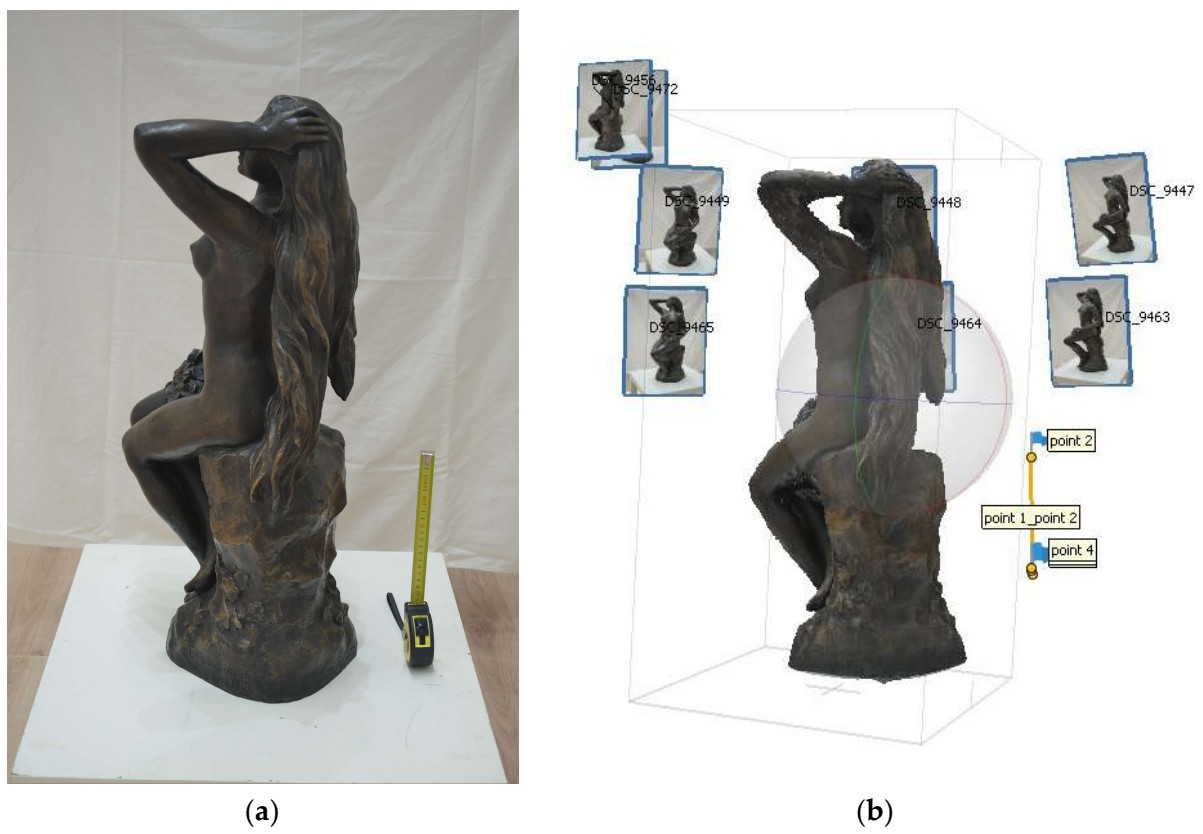

(**a**)                    (**b**)

**Figure 5.** Display of the ruler used for scaling the obtained 3D models to real dimensions in software: (**a**) sculpture on-site surveying with ruler; (**b**) obtained 3D model with added scale.

*3.2. Digitization of Drawings by 18th- and 19th-Century Artists and Sketchbook of Sava Šumanović*

The requirements for photographing the drawings by 18th- and 19th-century artists and the sketchbook of Sava Šumanović differ from the usual photogrammetric process. Before the actual photography, there is no need for any calculation, while it is necessary to take into consideration the lighting conditions of the environment. It is usually recommended to take photos outside so that there is no artificial lighting in the photos and to take them when it is cloudy so that there is no direct sunlight in the photos. If weather conditions are not ideal and there is direct sunlight, it should be taken care that it neither goes directly towards the camera nor that it is located behind it, but preferably that the light source is located to the left or right side of the camera. The works of art are placed on a specific stand, and the camera is positioned so that its optical axis is perpendicular to the stand at a distance such that it encompasses the entire image. Depending on the size of the image, the camera can move away or closer or take a photo from the same place while changing the focal length. Also, it is necessary to take into consideration if the works of art cannot stand on a pedestal (because it is too high) and to place them on a horizontal surface which is at the appropriate height so that it allows the objects to be captured from above. After all the photos are taken (example shown in Figure 6), the redundant parts are cut off in one of the digital image processing programs, and finally, a sketch block is created, within which it is possible to browse pages interactively. Digital flipbook appearances such as the front page and page layout with additional options are shown in Figure 7.

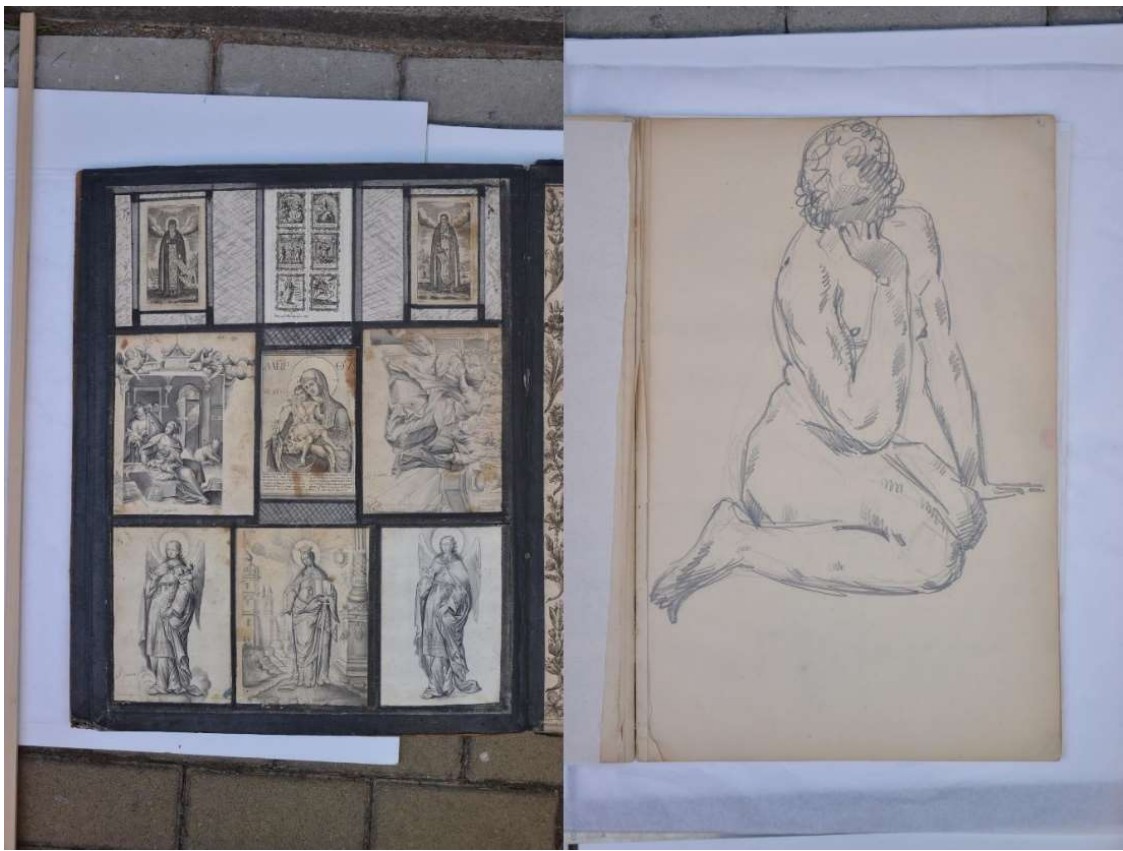

**Figure 6.** Example of digitized drawings by 18th- and 19th-century artists centuries and sketchbook of Sava Šumanović.

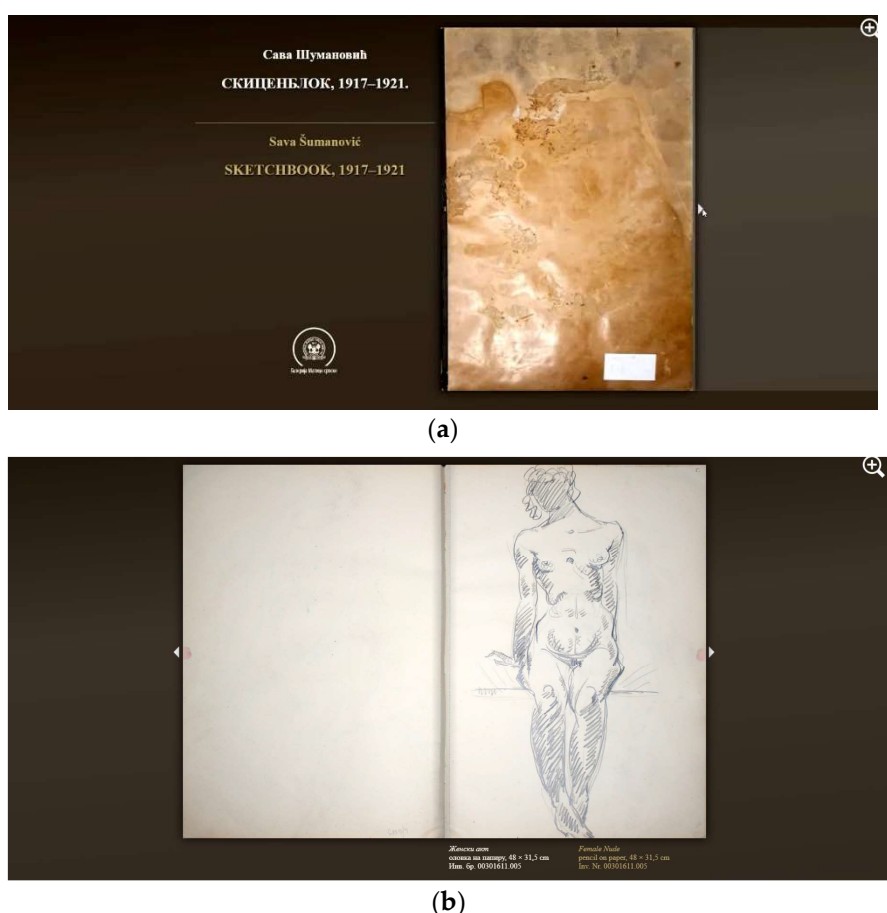

(**a**)

(**b**)

**Figure 7.** Digital flipbook appearance: (**a**) front page; (**b**) page layout with additional options.

### 3.3. Virtual Depot of the Gallery of Matica Srpska

In collaboration with the GMS, the created application is adjusted for non-immersive virtual reality systems, where the user has the opportunity to access the virtual art depot but also to find out more about the depot itself, to obtain instructions for navigating inside the virtual space, as well as to learn more about the project within which the creation of the depot was carried out. The main menu in the application is shown in Figure 8.

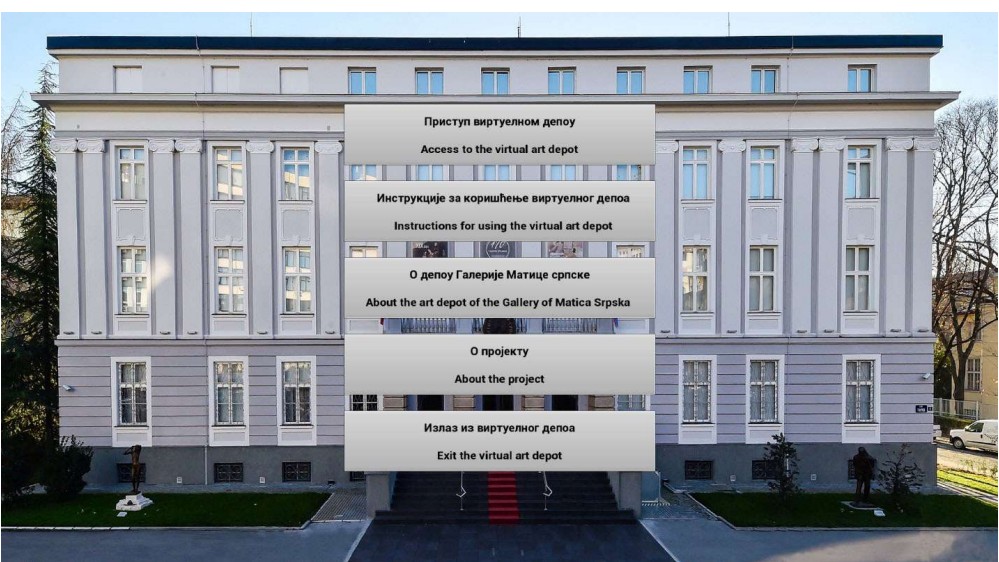

**Figure 8.** The main menu in the application offers the option to choose between Serbian and English.

The virtual space has the same dimensions as the depot in the real world, thus imitating its real-world role. In reality, only some employees have access to it, but in the virtual world, it is open to the public. Part of the project related to the creation of the application was realized in the Unreal Engine 4.26 [76] software and has the potential to be used with VR equipment, for which it is necessary to create a new application.

When selecting a button in the main menu, the user has the opportunity to find out how he/she can interact within the virtual depot. The user can move freely through it, with the possibility of moving the panels in the horizontal direction, as well as pulling the drawers out and in. On the panels, as well as in the drawers, there are works of art that correspond to their positions in the real world, and in this way, they are accessible to the public without the fear that they will be damaged, lost, or harmed in any way. In Figure 9, it can be seen the instructions for moving through the virtual depot written in Serbian and English.

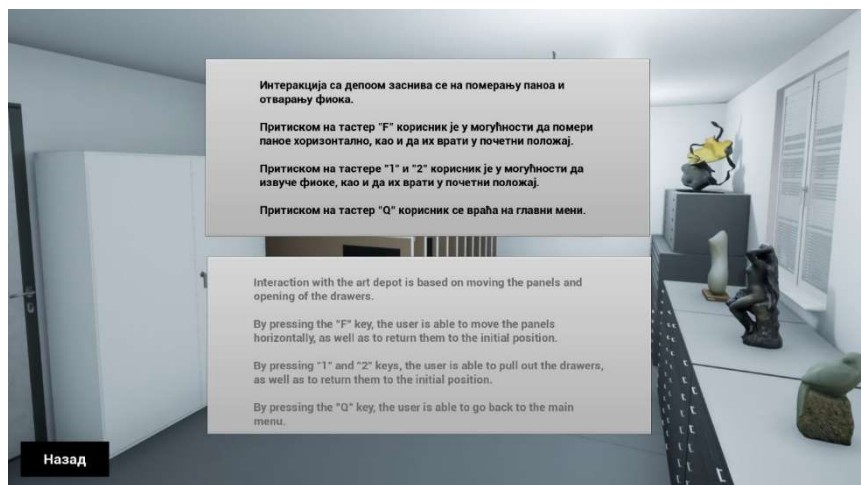

**Figure 9.** Instructions for using the virtual art depot in Serbian and English.

By pressing the button "About the art depot of the Gallery of Matica Srpska", the user of the application can find out more about the depot itself (see Figure 10).

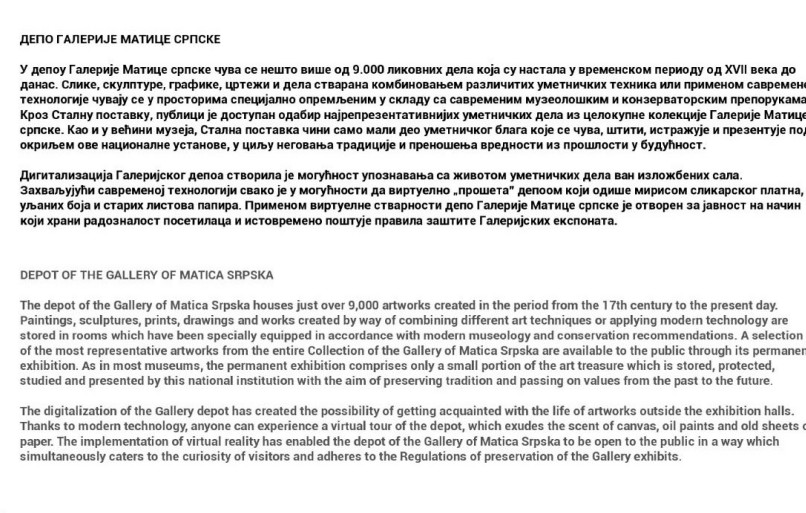

**Figure 10.** About the art depot of the Gallery of Matica Srpska.

The different segments of the virtual depot are shown in the Figures 11–14.

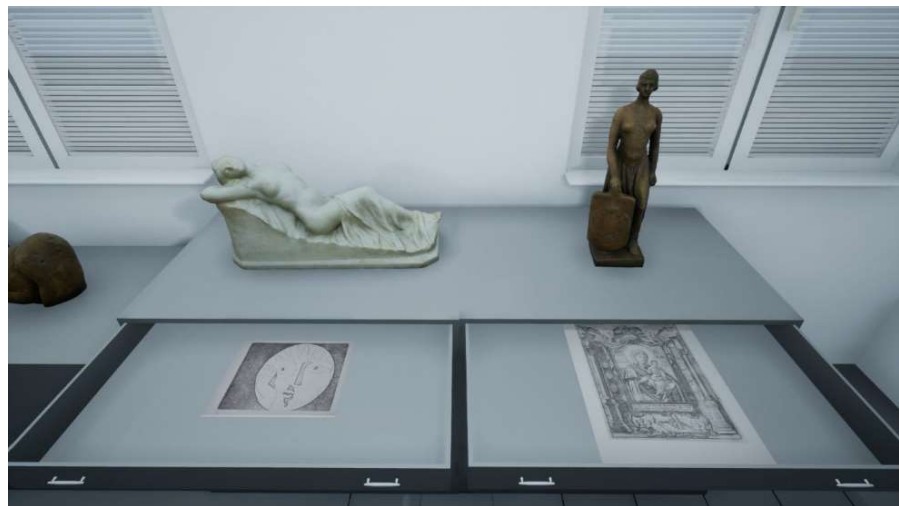

**Figure 11.** Showing the virtual depot segment (opening the drawer of the first chest of drawers).

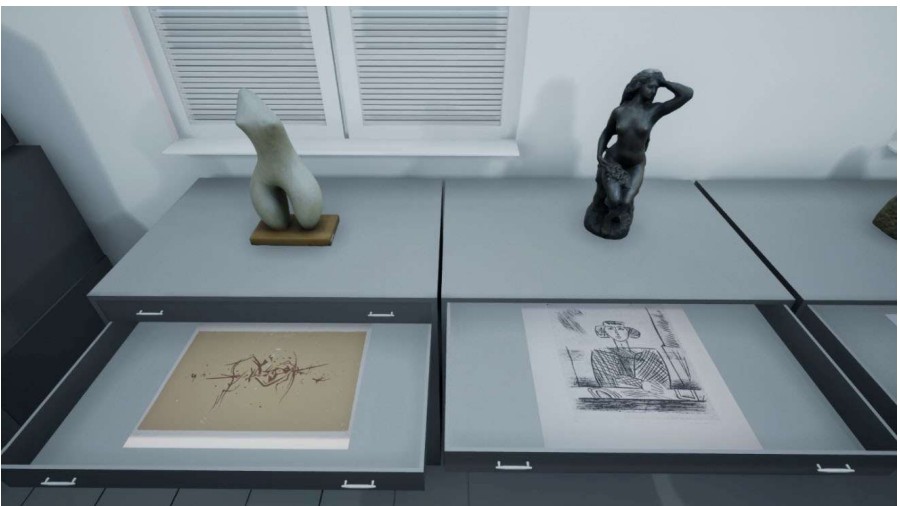

**Figure 12.** Showing the virtual depot segment (opening the drawer of the second chest of drawers).

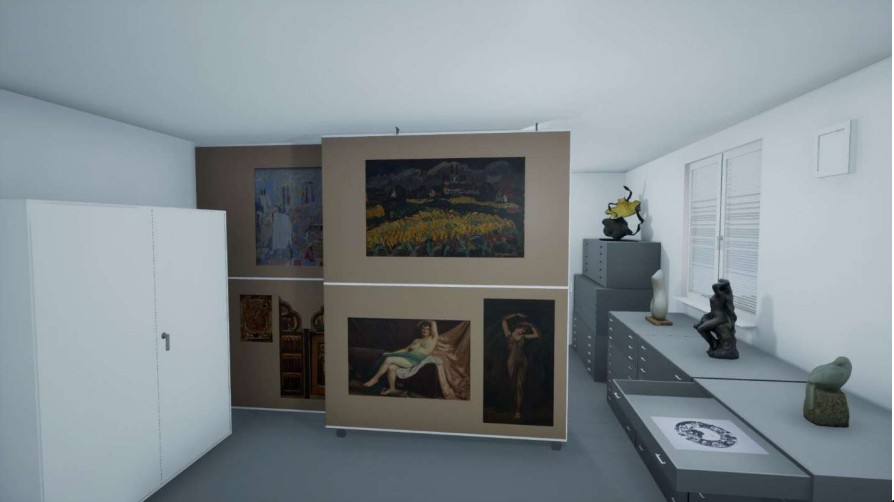

**Figure 13.** Displaying the virtual depot segment (panel scrolling)—left side of the room.

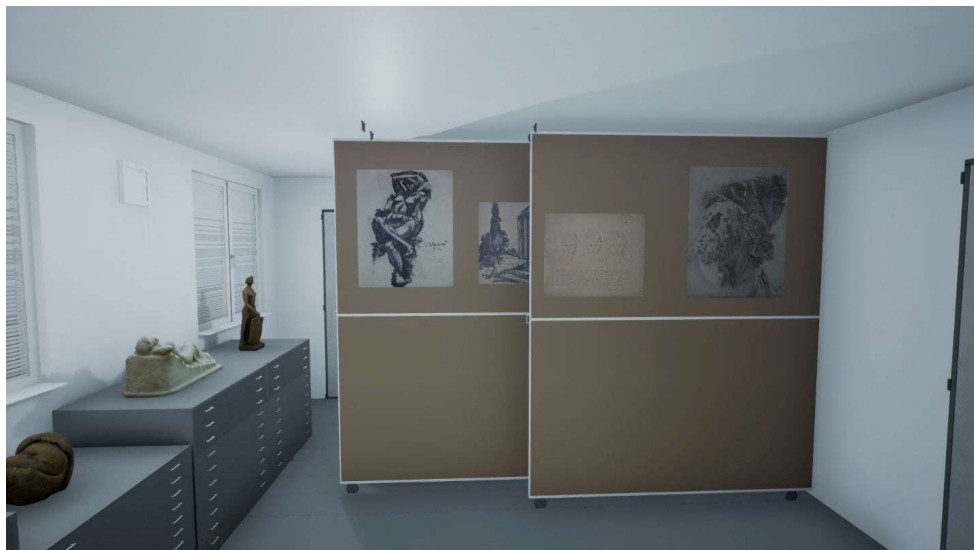

**Figure 14.** Displaying the virtual depot segment (panel scrolling)—right side of the room.

The main menu item transfers the user to the virtual space of the depot, within which the user can use the keyboard and mouse, as well as the commands described in more detail in the instructions segment.

The interactions based on moving panels and pulling out drawers are related to the created blueprint, which is adapted to each model. This means that there is a certain box collision, and at the moment when the intersection of the character and that box occurs, the input is enabled for any further interaction to take place. At that moment, the user can press a specific button in order to play a short animation, and that animation refers to the movement of a panel or drawer. The animation is the same for both geometries and is based on the vector displacement of the geometry in the length direction of the panel as if it has been pulled out by a hand. The initial phase of the animation refers to zero seconds, as well as zero displacement, and the end phase of the animation refers to two seconds, as well as to a certain distance in centimeters. This means that the panels and drawers cross a certain path along the task coordinate axis for a time interval of two seconds. Returning to the initial position is realized by playing the animation backward at the moment when the character presses the same button, regardless of how much the geometry has moved from the initial position at that moment.

In order for everything to work, it is necessary to use a static mesh inside each blueprint, which represents a specific 3D model. Therefore, the panel represents one static mesh, to which the dials are added as the second mesh, the frame as the third, and the artwork as the fourth. All models are grouped into one, so they all move together.

The idea of having a collision is to prevent the user from accessing some places, and thus, the collision can represent an invisible wall or shape through which the user cannot pass. Accordingly, it is necessary to take care of the drawers' collision because the user stands in front of the chest of drawers, and by pressing the button and pulling some of them out, they hit the user. And if they have a collision, two collisions can collide. In this case, the idea is not that the drawer blocks the character's ability to move, and their collision is excluded. As with the panels, each drawer is a separate static mesh, as is each artwork, so it was necessary to group everything into one model and to remove the collision in order to make a player move freely, even though the drawers are open. There are simple collisions in each panel to prevent the user from going through each one of them. Setting up a collision depends on a case-by-case basis, and the main thing to be guided by is the potential space for the user's movement. If some of the 3D models get in the way, it is possible to exclude the collision with those models.

## 4. Discussion

With the GMS virtual depot application, users have the opportunity to familiarize themselves with the Gallery's depot and its content, which is not open to the public for visits. As well as learning about the objects that are in the depot of the Gallery of Matica Srpska, it is also possible to learn about the ways of storing works of art and the conditions of keeping them. Users can see that the depot is organized in such a way as to ensure safe and inspected accommodation of museum artifacts relative to the type of material they are made of.

In the ever-evolving digital era, the digitization of cultural heritage has become a paramount endeavor for galleries and museums worldwide. However, the challenges associated with showcasing the vast and diverse collections stored in depots that are not accessible to the public have been a longstanding concern. This paper aims to highlight the significance of virtual presentations in virtual reality (VR) as a transformative solution for the digitization of these otherwise hidden treasures. By exploring the potential of VR technology, we can expand accessibility, enhance preservation efforts, and offer immersive experiences that bridge the gap between depots and the public. Depots are home to an extensive array of valuable artworks and historical artifacts that, due to limitations of space and preservation requirements, cannot be permanently displayed in galleries or museums. By using the virtual depot, certain artifacts that were not previously available to the public because they required special handling have become accessible to a wider audience, and in this way, the depot itself also gains importance.

Virtual reality offers a unique opportunity to overcome limitations by creating virtual presentations that transport users into a digital realm where they can explore these hidden collections. By using VR headsets or other similar devices, individuals can access and engage with these artifacts from anywhere in the world, breaking down geographical barriers and experiencing cultural heritage that was previously out of reach. Preserving delicate artworks and artifacts is of utmost importance in the realm of cultural heritage.

Traditional exhibition practices, while necessary for some pieces, expose objects to potentially damaging factors such as light, temperature fluctuations, and physical handling. VR technology allows for the creation of high-fidelity, interactive virtual environments that faithfully represent these objects without subjecting them to these risks. By digitizing depots through VR, institutions can ensure the long-term preservation of their collections while still providing an engaging and authentic experience to the public. The power of VR lies in its ability to create immersive and interactive experiences. By digitally reconstructing depots and incorporating realistic 3D models of objects, users can virtually explore these spaces and engage with the artifacts in unprecedented ways. Virtual reality opens the door for innovative educational experiences, enabling users to delve into the historical context, examine intricate details, and gain a deeper understanding of the cultural significance of each item. This technology has the potential to revolutionize the way people perceive and interact with cultural heritage, fostering curiosity, empathy, and appreciation for our shared history.

In many cases, gallery and museum depots hold cultural artifacts that are of immense importance to specific communities or regions. However, due to limited resources or geopolitical factors, access to these depots can be challenging or even restricted. Virtual reality offers a solution by digitally preserving and presenting these artifacts in an inclusive manner. By showcasing diverse cultural heritage in VR environments, we ensure that the richness and diversity of human history are celebrated and shared with audiences worldwide, fostering cultural understanding and promoting inclusivity. The integration of virtual reality into the digitization of gallery and museum depots presents a transformative solution for enhancing accessibility, preserving cultural heritage, and offering immersive experiences to the public. By leveraging the capabilities of VR technology, institutions can transcend physical limitations, reach broader audiences, and preserve the stories and significance of hidden treasures. The integration of virtual presentations in virtual reality is

not only an important advancement in the field of cultural heritage but also a crucial step towards a more inclusive and technologically empowered future.

The collection of the Gallery of Matica Srpska contains over 10,000 art objects. The Gallery is open to visitors from Tuesday to Friday, which means that if a visitor were to come to the Gallery every week on each of those days (no exceptions for holidays and non-working days), he/she would visit the Gallery sixteen times every month. Therefore, it can be concluded that if one artifact were examined in detail every day, it would take about 53 years to examine the GMS collection. Even if visitors were to look at several artifacts a day, it is assumed that some of the most interesting or larger ones cannot be fully understood with just one analysis, meaning that visitors would return to the GMS and look more than once. The virtual depot allows GMS visitors access to artworks that are not available to the public in real life, enabling them to view works of art of national importance much faster and in more detail, but, more importantly, they can do this at any opportunity and environment that suits them. In this sense, a potential visitor to the Gallery has the opportunity to virtually tour its depot on a smartphone or computer, for example, even while on a train or outside the borders of our country. This type of presentation also breaks down the physical barrier between the potential GMS's visitors and the artworks in terms of borrowing artworks and taking them home, which is not allowed due to the protection of the work from possible damage, but also copyright protection. When talking about copyright protection, it is taken into account that potential theft could happen in the form of the original work of art being taken out of the Gallery. It could be digitized, or a replica could be made using the same material and then be returned to the Gallery instead of the original artwork. Although the user of the Virtual Depot GMS application is fully immersed in the experience and feels as if actually walking through the GMS depot, at times he/she may feel as if he/she has taken home all the works of art he/she is viewing.

Since the GMS virtual depot application is available on the GMS website, there are data (measured via Google Analytics) that show the number of visits to the virtual depot of the Gallery of Matica Srpska, which are shown in Figure 15. The Figure shows data for the period from 1 January 2022 to 15 August 2023. The application was visited by only 283 users in about 592 days, which means that every second day, the application was seen by a new user on the website. The reason for such a small number of visits to the site is the insufficient presentation of the site and the ignorance of visitors that this option exists in GMS. In the following period, these applications should be made more visible, and therefore, the number of visitors should increase. The diagram shown in Figure 15 shows the number of visitors to the application in the specified period. Numbering 1–6 indicates that visitors from different countries searched; for example, under number 1, the search was carried out in the Serbian language, and it says Виртуелни депо|Галерија Матице српске, while number 5 shows the search in Greek. For each of the six items, it is shown how many times the application was viewed (first column), and by how many different users (second column), then the following columns show how much time was spent in the environment and how much money was earned. Since the application is free, GMS made RSD 0.

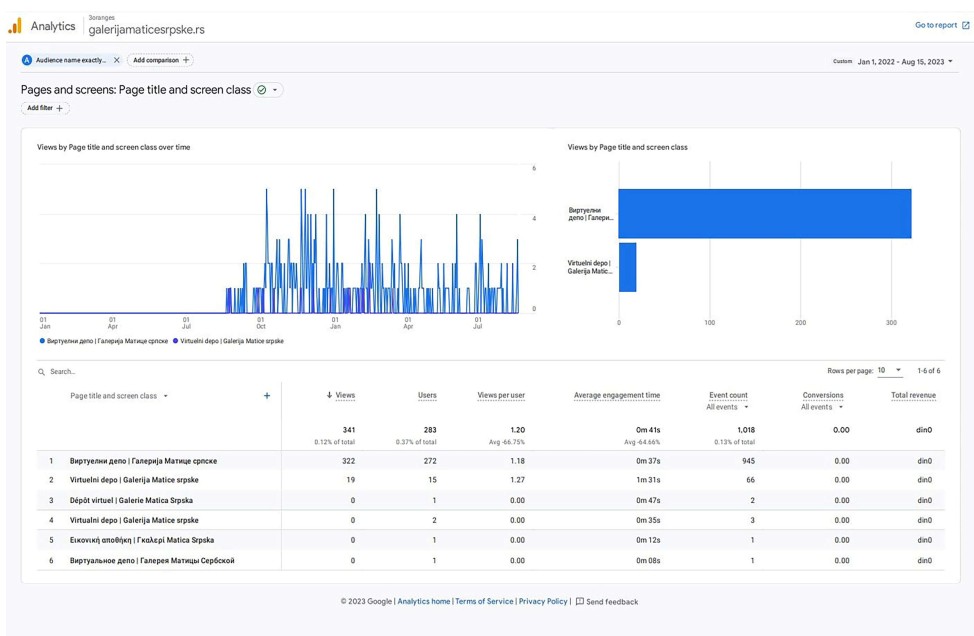

**Figure 15.** Statistics of visits to the virtual depot of the Galerija of Matica Srpska application.

*Technical Details*

As it was already mentioned, all sculptures were digitized using photogrammetry. Figure 16 shows a block diagram of the steps performed during the digitization procedure. It can be seen in Figure 16 that there are both pre-processing and post-processing steps. In the phase that precedes the actual photogrammetric surveying and the collection of photographs of the object to be digitized, it is necessary to create a surveying plan that defines in detail all the positions and orientations of the camera during the surveying. The phase of photogrammetric reconstruction of the object based on the images and the creation of a 3D model is an automatic procedure. It is necessary to import the collected images of the object into specialized software. Most often, a mistake made during surveying occurs or is noticed only after the first version of the model is obtained, i.e., after a generation of the sparse point cloud, which is the result of aligning the photos. If an error occurs, the photo-alignment step is repeated with additional settings or the photogrammetric surveying is completely repeated. If there is no error, the procedure continues until the final step, the result of which is a high-detail 3D model with a realistic, high-resolution texture. This is followed by post-processing, in which the number of polygons is reduced, and triangular polygons are converted into quadrilaterals. This step is performed to create a 3D model which is suitable for use in virtual reality applications. The goal is to optimize a model that will preserve as much detail as the one obtained by photogrammetry but will not be as demanding for rendering in real-time in VR applications. That is, it will not increase the rendering time, and therefore, there will be no problem when refreshing the image. For each of the sculptures, between three and four hours were necessary for photogrammetric surveying, while creating an individual surveying plan requires a day per sculpture. Generating a 3D model based on images requires one working day, while two working days are required for the optimization and retopology of each individual 3D model and its preparation for a VR application. Seven sculptures were digitized in this case study. In this way, larger objects such as monuments, iconostasis, parts of buildings, cathedrals, and facades of buildings of national significance can be digitized. For some of the mentioned types of objects, it may take twice as much time to generate a 3D model of a sculpture, e.g., for a cathedral or an iconostasis. Also, movable cultural heritage can be digitized in this way, e.g., coins, pottery, and shields. Digitizing these takes half the time it takes to digitize a sculpture. These objects can also be digitized by 3D scanning.

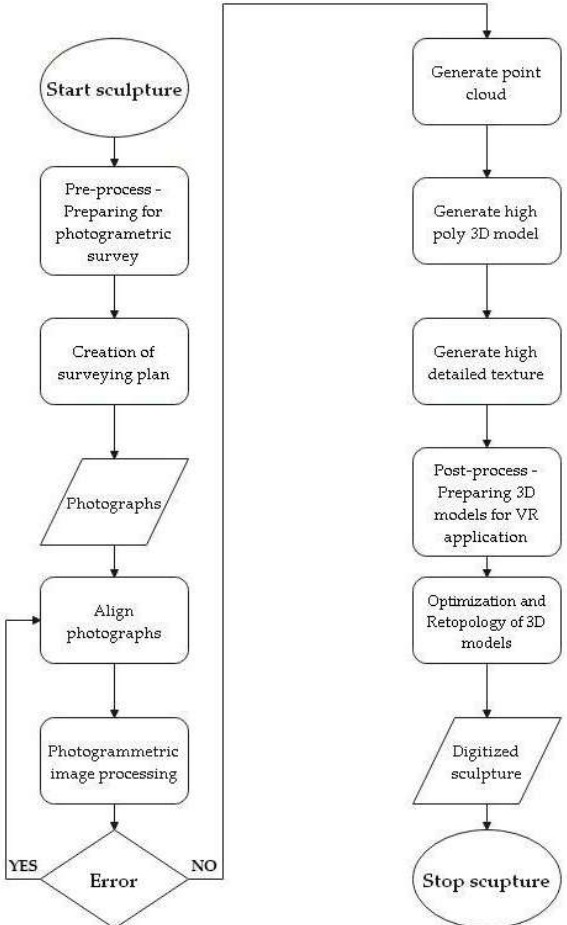

**Figure 16.** Block diagram of the photogrammetric workflow.

Figure 17 shows a block diagram of the steps that were implemented for the purpose of digitizing graphics and sketches. It can be seen in Figure 17 that the key steps in this procedure are the correct positioning of the drawings to be digitized (on the horizontal plane), the position of the camera whose optical axis is perpendicular to the plane on which the drawings are located, as well as the parameters of the camera, and in accordance with the lighting on the field. An error that can occur during surveying, which is noticed only when the captured photographs are analyzed, is that the positions of the lighting sources have changed on the field and that the parameters set during the surveying become incorrect in one moment. So, a new setting of the camera parameters is necessary. Also, it can happen that even though the lighting position does not change, the parameters are not set well, so a subsequent adjustment is made, and the surveying is repeated. After obtaining a set of photographs that are good, which means that there are no distortions or shadows on them and that the lighting was diffused when taken, redundant parts cutting off is performed. Redundant parts are the parts of the background, i.e., the visible part of the stand on which the drawings were located during photography or the frames in which the drawings are located, which should not be visible in the digital form of these drawings. In this way, and for the needs of the virtualization of the Gallery of the Matica Srpska depot, a total of 33 drawings were digitized. The time spent on capturing is between three and four hours. It took two working days to remove the redundant parts from the digitized drawings and prepare them for use in the Flipbook app that was created. In this way, it is possible to digitize books and all other types of documents of national importance, i.e., manuscripts. Also, in this way, it is possible to digitize fragments of folk costumes to preserve the scheme according to which the patterns on the parts of the clothing were created. If the patterns on

folk costumes need to be digitized as 3D objects, either photogrammetry or 3D scanning would be used for this purpose.

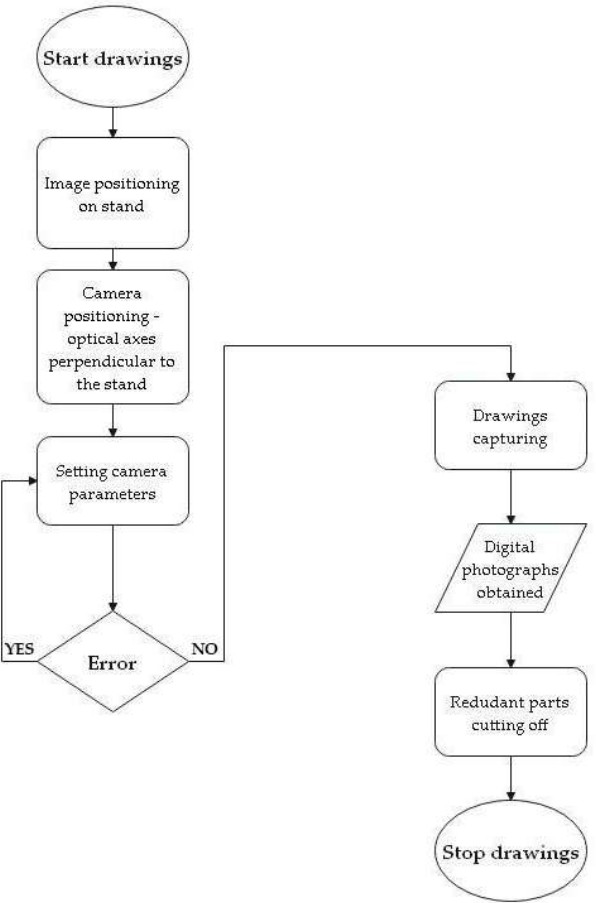

**Figure 17.** Block diagram of the digitization of the drawings workflow.

After the digitization of all the necessary objects, the depot's VR application was created. When creating an application that virtualizes the Gallery of Matica Srpska depot, it is unnecessary to use a complex collision, but one should strive towards the optimization of the scene, whether it is the number of polygons or the way of creating the collision. A simple collision was used that follows the shapes of drawers and panels, making it impossible for the user to pass through them, and due to its simplicity, it does not affect the performance of the computer needed to run the application.

The presented methodology allows sculptures to be simultaneously displayed in the virtual depot, then digitized drawings that are placed on panels, as well as drawings that are in drawers or viewed through a flip book presentation. In this way, the procedure was given that was applied to a certain number of exhibits (seven sculptures and 33 drawings) and in the project that was experimental and with a modest/small budget. Now, using this methodology, it is possible to digitize and display in VR the entire collection, numbering 10,000 works of art, but it requires both times, engaged personnel, and the provision of funds. Also, the sculptures were digitized using the Structure-from-Motion (SFM) photogrammetry, but protection was built into them, and according to the authors' previously shown procedure [59], virtual 3D models are also protected from abuse. Therefore, a team of authors responded to the request of GMS to show the real situation from the depot with acceptable technologies that do not damage works of art but are also financially acceptable techniques because their price is relatively low, and the results can be obtained relatively quickly. The originality of this work is that it shows what techniques should be used in

order to digitize all works of art of different forms that are in the GMS depot. This approach can also be applied to other depots, which store similar types of works of art.

With this project, 40 objects (seven sculptures and 33 drawings and graphics with the following information) from the GMS depot were finally made visible and accessible to the public. In particular, for the sketchbooks with drawings and graphics, it was the only way to make them available to the public because GMS visitors could not see them without touching them and flipping through sketchbooks. This means that digitized drawings and graphics can only be viewed virtually. If every visitor came into contact with the papers, their complete destruction or erasing of the drawings from the paper would occur very quickly. None of the seven digitized sculptures were available to the public as part of the GMS exhibitions. On the other hand, even if they were available within the exhibition, visitors should also not touch them in order to avoid possible damage, while in the virtual environment, it is possible to approach them closer and see them from all possible angles, i.e., in more detail.

## 5. Conclusions

In this paper, we presented the depot of the GMS by use of virtual reality (VR) experience showcasing photogrammetry digitized sculptures and sketchbooks from the GMS. The primary objective of this project was to leverage advanced imaging techniques to create accurate digital replicas of historical artifacts, allowing visitors to explore and interact with them in a virtual environment. Through photogrammetry surveying, we successfully captured and processed the data, generating high-quality 3D models of the sculptures, and we captured sketchbooks to obtain HD images of them. These digital replicas faithfully represent the intricate details and textures of the original artifacts, providing an immersive and realistic experience for users. The virtual experience of the GMS depot was designed considering factors such as user interface design, interaction mechanisms, and navigation. The integration of the 3D models into the VR environment was carefully executed to provide a seamless and engaging user experience. The virtual depot represents a significant advancement in the preservation and accessibility of cultural heritage. It offers numerous benefits, such as providing access to artifacts that may be fragile or inaccessible due to various constraints. Additionally, it allows for educational opportunities by providing contextual information and interactive elements within the VR experience.

To conclude, the virtual depot of the GMS, featuring photogrammetry digitized sculptures and sketchbooks, represents a groundbreaking approach to preserving and presenting cultural heritage. It demonstrates the potential of virtual reality to enhance accessibility, education, and engagement with art. The success and positive user feedback from this project encouraged further exploration and innovation in the field of digital heritage preservation and VR applications, ultimately contributing to the enrichment of cultural experiences for all.

By virtualizing the depot of the Gallery of Matica Srpska to increase its visibility and make it accessible to the public, seven sculptures became available to the public, as well as a total of 33 drawings, seven drawings that are placed in drawers and 25 pieces on panels. Currently, the number of digitized objects may not be large, but the idea (partially realized) of making the GMS depot available to the public has a great potential contribution. In fact, the methodology was created in a way that will guide the authors until the completion of the digitization of the depot in full. On the other hand, as already mentioned, the methodology applies to different types of objects and can be used as a guide for some new authors.

*Future Works*

While the Virtual Depot project has achieved considerable success, there are areas for further improvement and future work. Expanding the VR experience to include additional artifacts from the Gallery of Matica Srpska's collection would enhance the overall immersion and variety of content available. Moreover, integrating interactive educational

elements, such as historical context and artist biographies, would provide a more comprehensive learning experience for users. The information on how to store works of art in gallery depots in accordance with museum practice, regarding, for example, temperature and humidity, room lighting, and fire protection in the new version of the application, will become available to the public.

Furthermore, collaboration with other cultural institutions and sharing knowledge and expertise can contribute to the advancement of digital heritage preservation and virtual reality applications in the cultural sector. Additionally, developing a mobile VR application would extend the accessibility of the virtual depot to a wider audience, enabling users to engage with the collection using their smartphones or other portable VR devices.

## 6. Patents

In addition to the scientific contributions and advancements presented in this paper, the virtual depot VR application [77] may be eligible for patent protection due to its novel and inventive aspects. The integration of photogrammetry techniques with virtual reality technology to create an immersive and realistic experience for exploring digitized sculptures and sketchbooks from the GMS represents a unique and inventive approach. The method and system employed for capturing and processing data, generating high-quality 3D models, and seamlessly integrating them into a virtual environment demonstrate technical innovation. Furthermore, the potential embodiments and applications of the virtual depot, such as expanding the VR experience to include additional artifacts, incorporating interactive educational elements, and developing a mobile VR application, add further depth to its patentability. But, for now, the procedure for integrating photogrammetry 3D models into a VR environment described in this and our previous research [68] can be applied for different uses (gaming, virtual museums, virtual exhibitions, education, etc.).

**Author Contributions:** Conceptualization, R.O., D.I., M.O., I.V. and S.M.; methodology, R.O., D.I., M.O. and I.V.; software, M.O. and I.V.; validation, R.O. and D.I.; formal analysis, S.M.; investigation, R.O., D.I., M.O. and I.V.; resources, S.M.; data curation, S.M.; writing—original draft preparation, M.O., I.V. and S.M.; writing—review and editing, R.O. and D.I.; visualization, R.O., D.I., M.O. and I.V.; supervision, R.O. and D.I.; project administration, R.O.; funding acquisition, R.O. All authors have read and agreed to the published version of the manuscript.

**Funding:** This research received no external funding.

**Data Availability Statement:** The data presented in this study are openly available at https://www.galerijamaticesrpske.rs/virtuelni-depo/ (accessed on 2 July 2023).

**Acknowledgments:** The authors would like to thank the Gallery of Matica Srpska for the invitation to collaborate on projects in the field of digitization of cultural heritage and for working on the introduction of new technologies into the operations of the Gallery.

**Conflicts of Interest:** The authors declare no conflict of interest.

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
