# Peer review of "The Methodology of Virtualizing Sculptures and Drawings: A Case Study of the Virtual Depot of the Gallery of Matica Srpska"

_electronics, doi:10.3390/electronics12194157_

Round 1
Reviewer 1 Report
Dear Authors,
Your paper exhibits relevance and adherence to scientific writing norms, although it necessitates minor revisions to elevate its overall quality. My observations and recommendations are outlined below:
1-The abstract requires a succinct depiction of the employed methodology, encompassing approaches and techniques.
2-While the introduction section is long and solid, it would benefit from expansion. This expansion should encompass previous research in the subject area, a clear identification of the research gap, and a distinct outline of the paper's structure.
3-The literature review is presently sufficient when benchmarked against prior studies. An explicit definition of cultural treasures remains absent.
4- In discussion section, a more comprehensive analysis of the acquired results is necessary, along with pertinent comparisons.
5-The absence of a practical and social implications are a notable omission and should be addressed.
6-Ensuring adherence to proper referencing formats and inclusion of DOIs.
Author Response
We want to thank the Reviewer for the valuable comments and suggestions, which helped us to improve the quality of the manuscript.
Please note that the line numbers correspond to the PDF version of the revised manuscript without the Track Changes option.
We have addressed all comments in the revised version of the manuscript. Please follow our detailed responses to the comments below:
Reviewer comment:
Dear Authors,
Your paper exhibits relevance and adherence to scientific writing norms, although it necessitates minor revisions to elevate its overall quality. My observations and recommendations are outlined below:
1-The abstract requires a succinct depiction of the employed methodology, encompassing approaches and techniques.
Response:
1-In the revised version of the manuscript, methods and techniques which were used for digitization and virtual presentation of objects and drawings are listed. Please see lines 13-16.
2-While the introduction section is long and solid, it would benefit from expansion. This expansion should encompass previous research in the subject area, a clear identification of the research gap, and a distinct outline of the paper's structure.
Response:
2- In the revised version of the manuscript, the Introduction has been refined, and the difference between the author's project and projects from the used literature is highlighted. Please see lines 191-221. At the end of the Introduction (lines 286-290), in the revised version of the manuscript paper structure was added.
3-The literature review is presently sufficient when benchmarked against prior studies. An explicit definition of cultural treasures remains absent.
Response:
3-In the revised version of the manuscript definition of the cultural treasure (cultural heritage) was added according to UNESCO. Please see lines 85-92.
4- In discussion section, a more comprehensive analysis of the acquired results is necessary, along with pertinent comparisons.
Response:
4-More comprehensive analysis of the acquired results was added to the Discussion in the revised version of the manuscript. Please see lines 692-708.
5-The absence of a practical and social implications are a notable omission and should be addressed.
Response:
5- A practical and social implications were addressed in the Discussion in the revised version of the manuscript. Please see lines 709-720.
6-Ensuring adherence to proper referencing formats and inclusion of DOIs.
Response:
6-All references were checked. For some of them, it was not possible to find the DOI number, so was added link from where they were retrieved. Please see References in the revised version of the manuscript.

Reviewer 2 Report
The MS describe the digitization of seven sculptures and 33 drawing at the Gallery of Matica Srpska. I was a little puzzled as to why the MS was placed with Electronics rather than a MDPI journal such as Heritage, but this seems an editorial matter.
My main problem with the MS is the novelty and a confusion over the aims of the work, especially as it is such a small project at a time when other museums and libraries have virtual collections of tens of thousands of objects. Virtual reality and virtual presentations in museums are widely used, so I find them quite common and hard to sense exactly what is new here. The aims of the work as outlined in the MS seem wide ranging:
Line 13 "aim of this paper is to introduce the public to the virtual depot of the Gallery of Matica" - This does not make much sense. Why would the public read this paper in MDPI Electronics... surely they will go to other sources (in Serbian?) to find about the virtual depot
Line 139 "guidelines for the development of VR systems for cultural heritage." Was this an aim, although I wasn't sure that guidelines were ultimately presented.
Line 139 This paper aims to highlight the significance of virtual presentations in virtual reality (VR) as a transformative solution for the digitization of these otherwise hidden treasures.
I think it is important to be clear about the aims at the end of the introduction. It would also be useful to explain there exactly what is novel. This is especially a problem because "novel approach" is used as a keyword, but the novelty is never very clear.
TITLE The Methodology of Virtualizing Cultural Treasures: A Case Study of the Gallery of Matica Srpska seems reasonable -although I wondered if it would be more specific to replace Cultural Treasures: by sculpture and drawings. Does the title capture what is important about the paper, so I wondered why the virtual depot was missing, but this may be my fault as I was confused about the aims.
INTRODUCTION
Quite lengthy, probably OK for the audience, but no novel hypothesis orresearch aims presented at end of introduction, which would be a normal expectation .
Line 335 2.3. Virtual depot of the Gallery of Matica Srpska - - this seems about presentation rather than digitization so is a jump. from the ideas of 2.1 and 2.2. However, I wondered if this virtual depot was the main point of the paper. Is digital depot as a virtual reality tool the novel aspect. Most digital repositories I have used are not presented as virtual reality, but the authors would need to show way virtual reality is so important. As a researcher I am usually happy with the simple forms of presentation, but did the public users prefer a virtual reality approach?
Line 530 Figure 14 shows a block diagram of the steps that were implemented for the purpose of digitizing graphics and sketches. ... Is this a novel approach being shown in the figure or are these diagrams meant to represented the guidelines noted in Line 139.
Line 583 "It demonstrates the potential of virtual reality to enhance accessibility, education, and engagement with art." I don't believe this was well demonstrated as so few people used the system. Unfortunately the low usage is not explored very thoroughly. To me it was hardly surprising given the small size to the collection. Was feedback collected from those who visited the site, which could give a clue to poor uptake.
Line 607 application would extend the accessibility of the virtual depot to a wider audience, ... How does your study show this, when there were so few users?
In general understandable, but at time more thought could be used in word choice. As an example Line 212 "In recent times, virtually all museums have recognized the importance of digital communication in promoting " yet in the same paragraph the word "virtual" is used in relation to virtual reality . Using a word in two senses in the same paragraph is not a good idea.
Author Response
We want to thank the Reviewer for the valuable comments and suggestions, which helped us to improve the quality of the manuscript.
Please note that the line numbers correspond to the PDF version of the revised manuscript without the Track Changes option.
We have addressed all comments in the revised version of the manuscript. Please follow our detailed responses to the comments below:
Reviewer comment:
The MS describe the digitization of seven sculptures and 33 drawing at the Gallery of Matica Srpska. I was a little puzzled as to why the MS was placed with Electronics rather than a MDPI journal such as Heritage, but this seems an editorial matter.
Response:
We want to thank the Reviewer for the valuable comment, but we deal with Computer Graphics. We sent the manuscript to the Electronics journal in the Special Issue "Electronics and Computer Science for Cultural Heritage: Advancements, Preservation, and Applications", and there are topics such as: 3. Augmented reality (AR) and virtual reality (VR) applications for cultural heritage; 8. Human-computer interaction (HCI) and user interfaces for cultural heritage experiences.
Reviewer comment:
My main problem with the MS is the novelty and a confusion over the aims of the work, especially as it is such a small project at a time when other museums and libraries have virtual collections of tens of thousands of objects. Virtual reality and virtual presentations in museums are widely used, so I find them quite common and hard to sense exactly what is new here. The aims of the work as outlined in the MS seem wide ranging:
Response:
In this paper, the emphasis is not placed on the virtual presentation, because the depot itself is not used for exhibitions. A virtual depot differs from a virtual collection because it is a virtual representation of a real depot with works of art that are located there, stored in specific conditions and not accessible to the public. In the revised version of the manuscript, in the Introduction it was explained why only 40 works of art were digitized, and the reason was human resources and the available budget. Please see lines 216-221.
Reviewer comment:
Line 13 "aim of this paper is to introduce the public to the virtual depot of the Gallery of Matica" - This does not make much sense. Why would the public read this paper in MDPI Electronics... surely they will go to other sources (in Serbian?) to find about the virtual depot
Response:
There is no information available about the virtual depot accessible to the public, as it is only used to store artworks, and in this only way (by using our application) they could access it and find out more information about it.
Reviewer comment:
Line 139 "guidelines for the development of VR systems for cultural heritage." Was this an aim, although I wasn't sure that guidelines were ultimately presented.
Line 139 This paper aims to highlight the significance of virtual presentations in virtual reality (VR) as a transformative solution for the digitization of these otherwise hidden treasures.
Response:
Therefore, considering that there is no available information, on the website of the Gallery of Matica Srpska, users/visitors can access an application that represents a virtual depot, and the paper may popularize its existence. Guidelines for the development of VR systems are meant in the sense that the same form can be used for the presentation of other depots because it was not observed that there are presentations of something that is not normally available to the public.
Reviewer comment:
I think it is important to be clear about the aims at the end of the introduction. It would also be useful to explain there exactly what is novel. This is especially a problem because "novel approach" is used as a keyword, but the novelty is never very clear.
Response:
The new approach is to show the way in which works of art are stored, when they are not displayed in the exhibition space, and this data is not otherwise available to the public.
Reviewer comment:
TITLE The Methodology of Virtualizing Cultural Treasures: A Case Study of the Gallery of Matica Srpska seems reasonable -although I wondered if it would be more specific to replace Cultural Treasures: by sculpture and drawings. Does the title capture what is important about the paper, so I wondered why the virtual depot was missing, but this may be my fault as I was confused about the aims.
Response:
In the revised version of the manuscript the was changed to The Methodology of Virtualizing sculptures and drawings: A case study of the virtual depot of the Gallery of Matica Srpska, in that way we can cover a virtual depot in the title.
Reviewer comment:
INTRODUCTION
Quite lengthy, probably OK for the audience, but no novel hypothesis or research aims presented at end of introduction, which would be a normal expectation.
Response:
In the revised version of the manuscript, the Introduction has been refined and reorganized, also, the authors have tried to highlight the goals and hypotheses of the work more clearly.
Reviewer comment:
Line 335 2.3. Virtual depot of the Gallery of Matica Srpska - - this seems about presentation rather than digitization so is a jump. from the ideas of 2.1 and 2.2. However, I wondered if this virtual depot was the main point of the paper. Is digital depot as a virtual reality tool the novel aspect. Most digital repositories I have used are not presented as virtual reality, but the authors would need to show way virtual reality is so important. As a researcher I am usually happy with the simple forms of presentation, but did the public users prefer a virtual reality approach?
Response:
Public users do not have access to the depot at all, which is only available to employees of Matica Srpska Gallery, and they can only access it virtually, through the created application that represents that space, therefore the depot is the main point of work. Virtual reality and presentation of the depot of the Gallery of Matica Srpska in this way is a novelty for the audience in Serbia, because there are in our country, this kind of presentation of cultural heritage has not yet fully taken root.
Reviewer comment:
Line 530 Figure 14 shows a block diagram of the steps that were implemented for the purpose of digitizing graphics and sketches. ... Is this a novel approach being shown in the figure or are these diagrams meant to represented the guidelines noted in Line 139.
Response:
It is not a novel approach, it is an approach that has been used while digitizing graphics and sketches, summarized through a block diagram.
Reviewer comment:
Line 583 "It demonstrates the potential of virtual reality to enhance accessibility, education, and engagement with art." I don't believe this was well demonstrated as so few people used the system. Unfortunately the low usage is not explored very thoroughly. To me it was hardly surprising given the small size to the collection. Was feedback collected from those who visited the site, which could give a clue to poor uptake.
Response:
The potential is in the fact that there is a possibility for it to be available to the public, although it is not now, and the same workflow can be applied to applications with similar purposes. With greater work on the promotion of the application or on an even greater number of works of art, the number of visitors could increase proportionally. Please see lines 216-221 in the Introduction for an explanation of why the digitized collection is small.
Reviewer comment:
Line 607 application would extend the accessibility of the virtual depot to a wider audience, ... How does your study show this, when there were so few users?
Response:
It can be worked on the promotion of the application itself, which was not done much in the past period, but it was only placed on the Gallery of Matica Srpska website, which in itself may not have many visitors.
Reviewer comment:
Comments on the Quality of English Language
In general understandable, but at time more thought could be used in word choice. As an example Line 212 "In recent times, virtually all museums have recognized the importance of digital communication in promoting " yet in the same paragraph the word "virtual" is used in relation to virtual reality . Using a word in two senses in the same paragraph is not a good idea.
Response:
In the revised version of the manuscript, the English was proofread (file electronics-2598321_Track Changes_accepted.pdf).

Reviewer 3 Report
The topic of the study is new and interesting. The description of your work is accurate and well coincided. The main problems refer to the introduction. This part should be changed, starting from a description of the general topic related to the virtualization of cultural heritage, possibilities related to digital twins, meta verse, artificial intelligence, and so on. After un that, a focus on virtual reality for heritage visualization and lack in the literature (see paper on Digital twins in cultural heritage industry, or on the use of artificial intelligence, or digitalization for archeology https://doi.org/10.3390/su15043783, https://doi.org/10.3390/app11020870, https://doi.org/10.1016/j.patrec.2020.06.018). This last part is actually present in the paper, but it is not critic just a list of papers. Improve the critical value. Finally, the description of your museum must be inserted into a separate section as case study, not into the introduction. Structure of the paper lacks.
fine
Author Response
We want to thank the Reviewer for the valuable comments and suggestions, which helped us to improve the quality of the manuscript.
Please note that the line numbers correspond to the PDF version of the revised manuscript without the Track Changes option.
We have addressed all comments in the revised version of the manuscript. Please follow our detailed responses to the comments below:
Reviewer comment:
The topic of the study is new and interesting. The description of your work is accurate and well coincided. The main problems refer to the introduction. This part should be changed, starting from a description of the general topic related to the virtualization of cultural heritage, possibilities related to digital twins, meta verse, artificial intelligence, and so on. After un that, a focus on virtual reality for heritage visualization and lack in the literature (see paper on Digital twins in cultural heritage industry, or on the use of artificial intelligence, or digitalization for archeology https://doi.org/10.3390/su15043783, https://doi.org/10.3390/app11020870, https://doi.org/10.1016/j.patrec.2020.06.018). This last part is actually present in the paper, but it is not critic just a list of papers. Improve the critical value. Finally, the description of your museum must be inserted into a separate section as case study, not into the introduction. Structure of the paper lacks.
Response:
We would like to thank the Reviewer for the valuable comments and suggestions, which helped us to improve the quality of the manuscript. In the revised version of the manuscript, the description of our museum was inserted into a separate section as a case study, please see Section 2. All suggested papers: were analyzed and added to the Introduction and in References, please see lines 267-285 and References number 68-73.

Reviewer 4 Report
The paper elaborates on the methodology employed during the digitization process, encompassing data acquisition, processing, and visualization techniques of the collections of Gallery of Matica Srpska.
As indicated in the paper, to obtain quality results in the photogrammetric process it is necessary to have a set of photographs based on the detailed plan within a geodetic survey. This requires proper planning of the photo acquisition, as well as the use of appropriate data acquisition tools, software, and procedures.
Given the importance of the methodology used, it is noted only a brief description of the photogrammetric process followed has been given in the paper. Only examples of the original images and some of the photogrammetric products obtained are presented.
The methodology followed should be greatly improved. The following information must be presented in the paper, both in the text and including tables.
1. The equipment used for data acquisition (cameras). It is essential to present the metric characteristics of this equipment (focal length, sensor size, and image size).
2. Location parameters of the images during the data acquisition process. Such as the arrangement of the camera concerning the photographed objects, the distance at which the pictures are taken, the overlap between the pictures, GSD (Ground Sample Distance) demanded or the lighting used. Camera parameters, GSD, and shooting distance are closely linked. The paper indicates the importance of the GSD but does not report on the values considered when planning the data collection or the final results after generating the models.
3. The procedures used to obtain the geometrical information of the objects to be reproduced, which is essential to be able to give them scale, must be described. In the case of using point coordinates on the objects, the procedure followed must be indicated. The document indicates that it is crucial to take the data within a detailed geodetic survey plan, but this plan is not presented. It is necessary to indicate how this has been done in the case presented.
4. Modeling using photogrammetric software, such as Agisoft Metasahpe, involves several steps including the orientation of the cameras and the optimization of the orientation process, which leads to the determination of different types of errors, which must be controlled in the process. It is necessary that the steps followed and the errors determined in each of the steps are described. Part of the quality information of the photogrammetric processes should be presented in tables.
5. The photogrammetric process generates a variety of products, which can be used in 3D reconstruction, such as meshes, dense point clouds, digital elevation models, or orthomosaics. The products generated and the formats used for their visualization should be indicated.
Author Response
Response to Reviewer 4 Comments
We want to thank the Reviewer for the valuable comments and suggestions, which helped us to improve the quality of the manuscript.
Please note that the line numbers correspond to the PDF version of the revised manuscript without the Track Changes option.
We have addressed all comments in the revised version of the manuscript. Please follow our detailed responses to the comments below:
Reviewer comment:
The paper elaborates on the methodology employed during the digitization process, encompassing data acquisition, processing, and visualization techniques of the collections of Gallery of Matica Srpska.
As indicated in the paper, to obtain quality results in the photogrammetric process it is necessary to have a set of photographs based on the detailed plan within a geodetic survey. This requires proper planning of the photo acquisition, as well as the use of appropriate data acquisition tools, software, and procedures.
Given the importance of the methodology used, it is noted only a brief description of the photogrammetric process followed has been given in the paper. Only examples of the original images and some of the photogrammetric products obtained are presented.
The methodology followed should be greatly improved. The following information must be presented in the paper, both in the text and including tables.
- The equipment used for data acquisition (cameras). It is essential to present the metric characteristics of this equipment (focal length, sensor size, and image size).
Response:
In the revised version of the manuscript, it was added what equipment was used to survey with the specified camera characteristics. Please see lines 361-364.
- Location parameters of the images during the data acquisition process. Such as the arrangement of the camera concerning the photographed objects, the distance at which the pictures are taken, the overlap between the pictures, GSD (Ground Sample Distance) demanded or the lighting used. Camera parameters, GSD, and shooting distance are closely linked. The paper indicates the importance of the GSD but does not report on the values considered when planning the data collection or the final results after generating the models.
Response:
In the revised version of the manuscript, all parameters used for photogrammetric surveying are listed in the text and in the Table 1. Please see lines 365-374 and Table 1.
- The procedures used to obtain the geometrical information of the objects to be reproduced, which is essential to be able to give them scale, must be described. In the case of using point coordinates on the objects, the procedure followed must be indicated. The document indicates that it is crucial to take the data within a detailed geodetic survey plan, but this plan is not presented. It is necessary to indicate how this has been done in the case presented.
Response:
A scale was inserted in all digitized objects; an example is shown in Figure 5 in the revised version of the manuscript. The revised version of the manuscript now has 17 Figures. Please see lines 425-429.
- Modeling using photogrammetric software, such as Agisoft Metasahpe, involves several steps including the orientation of the cameras and the optimization of the orientation process, which leads to the determination of different types of errors, which must be controlled in the process. It is necessary that the steps followed and the errors determined in each of the steps are described. Part of the quality information of the photogrammetric processes should be presented in tables.
Response:
In the revised version of the manuscript, the details of the 3D reconstruction in the AgiSoft Metashape software are given. Please see lines 388-415.
- The photogrammetric process generates a variety of products, which can be used in 3D reconstruction, such as meshes, dense point clouds, digital elevation models, or orthomosaics. The products generated and the formats used for their visualization should be indicated.
Response:
In the revised version of the manuscript, all information about the generated product by use of photogrammetric reconstruction was added. Please see lines 388-398 and Figure 4. Point clouds (sparse and dense) are not exported from the software and in that case, there is no word here about the format in which they are saved.

Reviewer 5 Report
Interesting application article. The digitized 3D objects, either photogrammetry or 3D scanning are useful for the purpose.
1) Abstract: add methods and techniques used to the project
2) Introduction: A recent works on 3D small objects and in general on this field is the following a sample that should be included:
Michail I. Stamatopoulos and Christos-Nikolaos Anagnostopoulos (2023) digital modelling of ceramic sherds by means of photogrammetry and macrophotography: uncertainty calculations and measurement errors SCIENTIFIC CULTURE, Vol. 9, No. 3, 73-87 D(2022)OI: 10.5281/zenodo.7978086.
Kravari,K., Emmanouloudis, D., Korka, E & Vlachopoulou, A (2022) the contribution of information technologies to the protection of world cultural and natural heritage monuments “the case of ancient philippi, greece” , SCIENTIFIC CULTURE, Vol. 8, No. 3, 169-178 DOI: 10.5281/zenodo.6640266.
Călin Neamţu, Daniela Popescu, Răzvan Mateescu, Liliana Suciu, Dan Hurgoiu (2014) ABOUT QUALITY AND PROPERTIES OF DIGITAL ARTIFACTS, Mediterranean Archaeology and Archaeometry, Vol. 14, No 4, pp.55-64
Damala, A., Hornecker, E, van der Vaart, M, van Dijk, D., Ruthven, I ( 2016) THE LOUPE: TANGIBLE AUGMENTED REALITYFOR LEARNING TO LOOK AT ANCIENT GREEK ART. Mediterranean Archaeology and Archaeometry, Vol. 16, No 5,(2016),pp. 73-85, .DOI:10.5281/zenodo.204970
Olga Zaitceva, Mikhail Vavulin, Andrey Pushkarev, Evgeny Vodyasov, (2016) PHOTOGRAMMETRY: FROM FIELD RECORDING TO MUSEUM PRESENTATION (TIMIRYAZEVO BURIAL SITE, WESTERN SIBERIA) Mediterranean Archaeology and Archaeometry, Vol.16, No 5,(2016), pp. 97-103. DOI:10.5281/zenodo.204982
Vayia V. Panagiotidis and Nikolaos Zacharias (2022) DIGITAL MYSTRAS: AN APPROACH TOWARDS UNDERSTANDING THE USE OF AN ARCHAEOLOGICAL SPACE. SCIENTIFIC CULTURE, Vol. 8, No. 3, . 89-103, DOI: 10.5281/zenodo.6640278
3)Do you insert scale in the digitized objects? Why not?
4) Fig 12: It needs explanation in text this map.
5)The information on how to store works of art in gallery depots in accordance with museum practice, regarding, for example, temperature and humidity, room lighting, and fire protection has become available to the public: explain where is this shown?
6) Define what photogrammetry was used right from the Abstract please.
Author Response
Response to Reviewer 5 Comments
We want to thank the Reviewer for the valuable comments and suggestions, which helped us to improve the quality of the manuscript.
Please note that the line numbers correspond to the PDF version of the revised manuscript without the Track Changes option.
We have addressed all comments in the revised version of the manuscript. Please follow our detailed responses to the comments below:
Reviewer comment:
Interesting application article. The digitized 3D objects, either photogrammetry or 3D scanning are useful for the purpose.
1) Abstract: add methods and techniques used to the project
Response:
1) In the revised version of the manuscript, methods and techniques which were used for digitization and virtual presentation of objects and drawings are listed. Please see lines 13-16.
2) Introduction: A recent works on 3D small objects and in general on this field is the following a sample that should be included:
Michail I. Stamatopoulos and Christos-Nikolaos Anagnostopoulos (2023) digital modelling of ceramic sherds by means of photogrammetry and macrophotography: uncertainty calculations and measurement errors SCIENTIFIC CULTURE, Vol. 9, No. 3, 73-87 D(2022)OI: 10.5281/zenodo.7978086.
Kravari,K., Emmanouloudis, D., Korka, E & Vlachopoulou, A (2022) the contribution of information technologies to the protection of world cultural and natural heritage monuments “the case of ancient philippi, greece” , SCIENTIFIC CULTURE, Vol. 8, No. 3, 169-178 DOI: 10.5281/zenodo.6640266.
Călin Neamţu, Daniela Popescu, Răzvan Mateescu, Liliana Suciu, Dan Hurgoiu (2014) ABOUT QUALITY AND PROPERTIES OF DIGITAL ARTIFACTS, Mediterranean Archaeology and Archaeometry, Vol. 14, No 4, pp.55-64
Damala, A., Hornecker, E, van der Vaart, M, van Dijk, D., Ruthven, I ( 2016) THE LOUPE: TANGIBLE AUGMENTED REALITYFOR LEARNING TO LOOK AT ANCIENT GREEK ART. Mediterranean Archaeology and Archaeometry, Vol. 16, No 5,(2016),pp. 73-85, .DOI:10.5281/zenodo.204970
Olga Zaitceva, Mikhail Vavulin, Andrey Pushkarev, Evgeny Vodyasov, (2016) PHOTOGRAMMETRY: FROM FIELD RECORDING TO MUSEUM PRESENTATION (TIMIRYAZEVO BURIAL SITE, WESTERN SIBERIA) Mediterranean Archaeology and Archaeometry, Vol.16, No 5,(2016), pp. 97-103. DOI:10.5281/zenodo.204982
Vayia V. Panagiotidis and Nikolaos Zacharias (2022) DIGITAL MYSTRAS: AN APPROACH TOWARDS UNDERSTANDING THE USE OF AN ARCHAEOLOGICAL SPACE. SCIENTIFIC CULTURE, Vol. 8, No. 3, . 89-103, DOI: 10.5281/zenodo.6640278
Response:
2) All mentioned literature were added in the Introduction of the revised version of the manuscript. Please see lines 190-224, and references 38-46.
3)Do you insert scale in the digitized objects? Why not?
Response:
3) A scale was inserted in all digitized objects; an example is shown in Figure 5 in the revised version of the manuscript. The revised version of the manuscript now has 17 Figures. Please see lines 425-429 and Figure 5.
4) Fig 12: It needs explanation in text this map.
Response:
4) In the revised version of the manuscript Figure 12 is Figure 15 (because of adding new Figures). An explanation was added for the diagram shown in Figure 15. Please see lines 613-621.
5)The information on how to store works of art in gallery depots in accordance with museum practice, regarding, for example, temperature and humidity, room lighting, and fire protection has become available to the public: explain where is this shown?
Response:
5) In this paper, the procedure of saving the works of art is not the subject of virtualization, but the work of art itself. In the Introduction we mentioned how to store works of art in gallery depots in accordance with museum practice, regarding temperature and humidity, room lighting, and fire protection has become available to the public, but for now, it cannot be seen in the application. The idea is to enable the visibility of the mentioned information in the improved version of the application.
6) Define what photogrammetry was used right from the Abstract please.
Response:
6) In the revised version of the manuscript, lines 14-15 (in the Abstract) states that it used Structure-from-Motion photogrammetry.

Round 2
Reviewer 2 Report
The authors have made a number of changes. I now understand the notion of digitising materials in a depository and understand why this work was done. It is less clear that this project will attract wide public interest, despite the value of the collection. However it is not necessarily the role of academic reviews to assess longer term public success of digitising these materials.
Copy editing should be able to cope with this...
Author Response
We would like to thank the Reviewer once again for his valuable comments and suggestions, which helped us to improve the quality of the manuscript. We will make sure that more people know about our GMS presentation in the near future.

Reviewer 4 Report
As indicated in the previous review, modeling using photogrammetric software, such as Agisoft Metasahpe, involves several steps, such as orienting the cameras or optimizing the orientation process. Errors made in the process are not indicated in the document. These errors give insight into the quality of the models obtained.
Author Response
In the revised version of the manuscript RMS error and maximal reprojection error values were added for every single sculpture, please see Table 2. Image Quality values are mentioned, too. Please see lines 411-417.

Reviewer 5 Report
Ony ref no 45 the journal is missing Scientific Culture. Also references are not consistent (some capital some small lettering.
Author Response
In the revised version of the manuscript, it was added Scientific Culture. Please see reference 45. The titles of some of papers are written in capital letters because they are listed that way in the original, but it has been corrected and now it is uniform. Please see references 7, 18, 26, 38 and 60, too.
